# Programmable RNA base editing with photoactivatable CRISPR-Cas13

Jeonghye Yu ®[1], Jongpil Shin[1], Jihwan Yu ®[1], Jihye Kim[1], Daseuli Yu ®[2] & Won Do Heo ®[1,3] ✉

CRISPR-Cas13 is widely used for programmable RNA interference, imaging, and editing. In this study, we develop a light-inducible Cas13 system called paCas13 by fusing Magnet with fragment pairs. The most effective split site, N351/C350, was identified and found to exhibit a low background and high inducibility. We observed significant light-induced perturbation of endogenous transcripts by paCas13. We further present a light-inducible base-editing system, herein called the padCas13 editor, by fusing ADAR2 to catalytically inactive paCas13 fragments. The padCas13 editor enabled reversible RNA editing under light and was effective in editing A-to-I and C-to-U RNA bases, targeting disease-relevant transcripts, and fine-tuning endogenous transcripts in mammalian cells in vitro. The padCas13 editor was also used to adjust post-translational modifications and demonstrated the ability to activate target transcripts in a mouse model in vivo. We therefore present a light-inducible RNA-modulating technique based on CRISPR-Cas13 that enables target RNAs to be diversely manipulated in vitro and in vivo, including through RNA degradation and base editing. The approach using the paCas13 system can be broadly applicable to manipulating RNA in various disease states and physiological processes, offering potential additional avenues for research and therapeutic development.

The development of class II, type VI bacterial clustered, regularly interspaced, short palindromic repeat CRISPR-Cas13 systems for mammalian cells has continued to expand the field of genome engineering[1–4]. The Cas13 nuclease can bind to specific RNAs; when the target RNA complements at least 20 to 30 nucleotides of a CRISPR RNA (crRNA), the Cas13-crRNA complex is activated and the higher eukaryotes and prokaryotes nucleotide-binding domain (HEPN domain) undertakes RNA cleavage[5]. The initially discovered Cas13 nuclease, *Leptotrichia wadei* Cas13a (LwaCas13a), could target specific RNAs with the nuclease-deactivated dead Cas protein (dCas); this allowed for the labeling of endogenous RNAs, even lncRNAs, in mammalian cells[1,2,6]. More recent work showed that *Prevotella sp. P5-125* Cas13b (PspCas13b) can edit specific RNAs using dCas protein fused with the catalytic domain of ADAR2 (adenosine deaminase acting on RNA type 2),

thereby expanding therapeutic strategies for disease-specific genome engineering[3].

Many smaller Cas13 systems have been developed and characterized with the goal of enabling the therapeutic delivery of RNA editing systems[7,8]. While these compact systems may be more easily packaged into adeno-associated virus (AAV) vectors, their lack of temporal control remains a significant limitation. The inherent dynamic and temporal nature of RNA means that its modulation must be precisely controlled, but such functionality is not offered by conventional Cas13 systems. Moreover, the continuous expression of untargeted regulatory proteins and Cas13 may lead to undesirable cellular effects. To address these limitations, the field needs inducible platforms that provide the temporal control needed to enable the development of next-generation RNA modulation techniques.

[1]Department of Biological Sciences, Korea Advanced Institute of Science and Technology (KAIST), Daejeon, Republic of Korea. [2]Life Science Research Institute, KAIST, Daejeon, Republic of Korea. [3]KAIST Institute for the BioCentury (KIB), KAIST, Daejeon, Republic of Korea. ✉e-mail: wondo@kaist.ac.kr

Recently introduced inducible Cas13 systems utilizing abscisic acid (ABA) and 4,5-dimethoxy-2-nitrobenzyl (DMNB)-caged ABA offers temporal and light control[9,10]. However, these inducers still rely on chemical treatment and thus require additional and complicated steps for in vivo application. The blue light-inducible system involves a much simpler induction process without additional chemicals. Moreover, blue light is a less cytotoxic stimulus than UV light, making it safer and more broadly applicable for in vitro and in vivo applications. Thus, by strategically splitting the Cas13 protein to reduce its size and incorporating light-inducible control, our system successfully addresses the current limitations that hamper the deliverability and precision of RNA targeting.

This study presents an inducible method for modulating RNA using a chemogenetic, split Cas13, and an optogenetic, photoactivatable Cas13 (paCas13); these systems are based on the FKBP-FRB dimerization domains and the Magnet system, respectively[11,12]. To generate inducibility, we split the PspCas13b protein using an in situ restructuring process applied by Alphafold2. This was necessary because, in contrast to the well-known Cas9 proteins, the protein structure was unavailable for mammalian cell-applied Cas13 proteins[13]. To reassemble the split Cas13 protein and thereby trigger RNA disturbance, we used rapamycin treatment to induce FKBP to interact with its partner FRB, or blue light to trigger the Magnet system. We further developed a light-inducible base editor called the padCas13 editor, which can enable RNA base editing with a nuclease-deactivated dCas13 protein in vitro and in vivo, as revealed by luciferase reporter activity.

In this work, the developed paCas13 system can be utilized to examine the function of specific RNAs and precisely control endogenous transcripts in living cells to significantly alter the target protein using light. The split fragments of paCas13 are capable of overcoming the limits of AAV-based delivery, increasing the applicability of this system for therapeutic purposes. Its potential to enable programmable epitranscriptome manipulation should facilitate future therapeutic applications.

## Results

### Design of chemically inducible split Cas13

We set out to generate an inducible system by splitting the Cas13 protein. From among the diverse orthologs of the CRISPR-Cas13 system, we selected PspCas13b, because it is known as the most efficient Cas13 protein for fluorescent labeling of specific target RNA, and exhibits better target specificity than other Cas13 proteins[6]. In addition, the crRNA of PspCas13b has a relatively long spacer sequence (at least 30 nucleotides) that could provide better specificity than Cas13 orthologs with shorter spacer sequences.

To split the protein, we needed to select solvent-accessible loops whose cleavage would not disturb any secondary structure element such as an α-helix or β-sheet (Fig. 1a). Because the three-dimensional structure of PspCas13b has not yet been revealed, we performed a deep learning-based prediction of the protein structure using AlphaFold2[13,14]. To improve the prediction and build a structure with a high per-residue confidence score called predicted local distance difference test (pLDDT) indicating reliable folding results, we conducted up to 12 additional recycling processes, as guided by performance improvements observed in previous research[13]. The average pLDDT score was 89.04, suggesting that there was a high confidence in the predicted structure. We further reconstructed functional domains with reference to the structure of *Prevotella buccae* Cas13b (PbuCas13b)[15], and then examined the final predicted protein structure to identify candidate split sites (Fig. 1b).

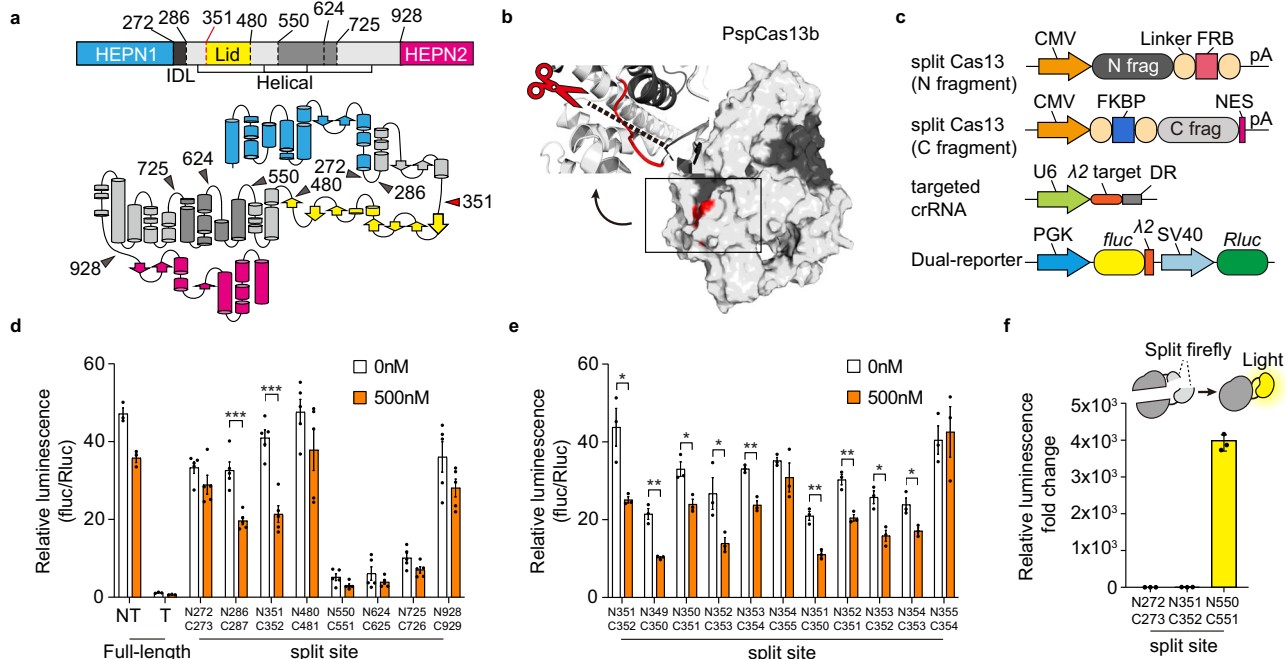

**Fig. 1 | Screening and characterization of split site on PspCas13b. a** Mapping the cleavage site onto the structure of the PspCas13 protein. The split sites are marked by arrows; the N-terminal HEPN1 is shown as blue, and the C-terminal HEPN2 is shown as magenta. IDL, inter-domain linker. **b** Predicted protein structure for the cleavage site of PspCas13b. Red mark indicate the split sites for split-Cas13-1 (N351/C352). The N-terminal fragment is shown as dark gray, and the C-terminal fragment is shown as gray. **c** Schematic of the construct designs for screening split sites of Cas13. All split Cas13 candidates were expressed under the CMV promoter, and the FKBP-FRB system was introduced into the split fragments. The crRNA targets the λ2 sequence in the dual-luciferase reporter plasmid. **d** Ligand-inducible split Cas13 activity, as measured by dual-luciferase assay performed after the addition of 500 nM rapamycin (*n* = 5 independent experiments). **e** Second screening of Cas13 split sites in the vicinity of residues 351 and 352 (*n* = 3 independent experiments). **f** Split-fluc reassembly assay examining the self-assembly activity of split Cas13 fragments without crRNA, as represented by a signal from the dual luciferase assay (*n* = 3 independent experiments). *P < 0.05, **P < 0.01, ***P < 0.001,****P < 0.0001 as determined by two-tailed Student's t-test. All error bars represent the mean ± s.e.m.

We chose eight candidate split sites based on their localization in loop regions and their amino acid properties, taking care to avoid the terminal HEPN domains so as to maintain the nuclease activity of Cas13 (Fig. 1a). To develop the chemically inducible system, which was used to test the candidates, we coupled each split fragment pair to the interaction partners of the rapamycin-inducible dimerization system, FKBP and FRB, using a relatively long GS linker to avoid steric hinderance of nuclease activity (Fig. 1c).

To investigate the efficiency of each candidate chemically inducible split Cas13 system, we conducted an inducibility test. To minimize well-to-well variations, we used a dual-luciferase assay comprising a single plasmid containing the *firefly* (fluc) and *Renilla* (Rluc) luciferase genes controlled by distinct promoters (Fig. 1c), and normalized luciferase activity as the fluc/Rluc ratio. The split Cas13 mainly functions in the cytosol due to the presence of a nuclear export signal (NES) tag, which mediates its transport from the nucleus. The cytosol is abundant with RNA-binding proteins that interact dynamically with RNAs. In an effort to improve the accuracy of our screening of split Cas13 candidates, we incorporated a λ2 bacteriophage sequence[16], which can be targeted for RNA degradation and is not present in human cells, into the 3′ UTR after the stop codon. This strategy aimed to minimize the potential for interference from endogenous RNA-binding proteins and thereby enhance the precision of our screening in terms of coding region translation and RNA degradation (Fig. 1c). We then applied rapamycin for 24 hours and assessed rapamycin-induced nuclease activity. Our results revealed that the split site N351/C352 could function effectively for rapamycin-inducible RNA degradation, while the split sites N550/C551, N624/C625, and N725/N726 exhibit background activity (Fig. 1d and Supplementary Table 1).

To characterize background activity, we used both experimental and computational approaches. We transfected individual dimerization domain-fused split Cas13 fragments with the targeted crRNA. We found that the background activity was unrelated to the residual nuclease activity of the split Cas13 fragments (Supplementary Fig. 1a). In addition, a split-firefly luciferase assay was conducted to clarify whether the fragments could spontaneously reconstitute to show activity[17]. From our previous screening under rapamycin conditions, we selected three distinct split sites based on their RNA degradation activity: N272/C273 with lower inducible activity, N351/C352 with ligand-inducible activity, and N550/C551 with spontaneous background activity. These experiments were conducted without crRNA to specifically avoid crRNA-dependent effects. Each split fragment was fused with split-firefly luciferase and both were expressed together in HEK 293 T cells. The results demonstrated that the split sites N272/C273 and N351/C352 with inducible activity exhibited no signal of firefly luciferase without dimerization domain, but the split site N550/C551 with background activity exhibited a luciferase signal (Fig. 1f). Therefore, we suggest that the background activity of split Cas13 fragments is caused by spontaneous reconstitution of split sites.

Second, to determine whether the structural interaction of split Cas13 could affect background activity, we analyzed the experimentally validated combinations and all candidate split fragments using AlphaFold2-based structural prediction[13]. Granted, AlphaFold2 is not fully reliable in predicting ribonucleoprotein structures, and Cas13 undergoes significant conformational changes upon crRNA binding[15]. However, AlphaFold2 is a valuable tool for bridging the gap between experimental and computational insights, such as in our experimental context. The surface interactions of these fragments were further investigated using MaSIF-site[18], which is a geometric deep learning-based tool that provides molecular surface interaction fingerprinting (Supplementary Fig. 2a). When combined, the results from the experimental and computational analyses suggested that there is a relationship between auto-assembly activity and the extent of the labeled interface. Specifically, N550/C551, N624/C625, and N725/C726 exhibited higher auto-assembly activity in the experimental results

(Fig. 1d) and displayed more extensive labeled interfaces and open structural interaction pockets in the computational analysis (Supplementary Fig. 2b, c). Notably, these high-activity combinations are located within the crRNA binding region of PspCas13b. This region exhibits elevated positive electrostatic potential, which may underlie the enhanced auto-assembly activity observed at the high-activity split sites (Supplementary Fig. 2d).

We further screened Cas13 split sites around amino acid 351 to identify the most effective split-site (Fig. 1e). Based on our collective results, we selected inducible split pairs, split Cas13-1 (N351/C352) and split Cas13-2 (N351/C350), for use in subsequent experiments. Thus, we herein developed a chemically inducible RNA degradation system that functions by reassembling Cas13 from split fragments, which appears to be the most effective strategy in terms of providing a low background activity and high inducibility.

### Development and characterization of a photoactivatable Cas13

After establishing a chemically inducible RNA degradation system by reassembling Cas13 from split fragments, we placed this system under optogenetic control. The protein sizes are similar between the rapamycin-inducible dimerization system and the Magnet system used for photoinducible dimerization, allowing us to smoothly transition to a light-controlled approach. To produce a light-inducible Cas13 system, we fused the Magnet proteins, which undergo photoinducible dimerization, with fragment pairs having the same configuration as our chemically inducible split Cas13 (Fig. 2a). The Magnet system consists of a positively charged Magnet (pMag) and a negatively charged Magnet (nMag), which quickly heterodimerize under light illumination[12]. There are several engineered Magnet systems based on kinetic mutations within the Per-Arnt-Sim (PAS) core[12]. Because previous studies of photoactivatable Cas9 utilized the combination of pMag and nMagHigh1 (which exhibit high affinity) and provided information on the photoswitchable properties of the nuclease[19], we used pMag and nMagHigh1 (nMagH) in developing our photoactivatable Cas13 system (named paCas13).

To validate the light-inducible dimerization of paCas13, we evaluated the target RNA knockdown activity using the dual-luciferase system and a light-emitting diode (LED) plate emitting 488-nm blue light. The split sites selected from the above-described experiments were assessed in the paCas13 system to identify whether its inducibility was consistent with that of the split Cas13 system. We transfected HEK 293 T cells with paCas13 fragments and target crRNA along with a dual-luciferase reporter containing the λ2 sequence, and then evaluated the light inducibility for 24 hours. We observed light-inducible RNA degradation mediated by reassembly of paCas13 and found that the luciferase activity patterns paralleled those obtained using the chemically inducible split Cas13 assay (Fig. 2b). To determine whether any fragment exhibited nuclease activity when expressed individually, we screened background activity by transfecting cells with vectors encoding each fragment, the crRNA, and a dual-luciferase reporter (Supplementary Fig. 1b). We found that background activity was independent of the remaining nuclease activity of each fragment. In accordance with the above results of the split Cas13, two light-inducible candidates were designated paCas13-1 (N351/C352) and −2 (N351/C350). The two differed by only one amino acid, but paCas13-1 had a lower background signal and paCas13-2 had greater RNA targeting activity, indicating that their light-inducible activity was substantially different (Fig. 2c). To determine the efficacy of RNA targeting with the paCas13, we performed a time-course assay. We observed that paCas13-2 displayed faster and more stable RNA degradation activity than paCas13-1 (Fig. 2d). This suggests that paCas13-1 and paCas13-2, which differ by a single amino acid, exhibit markedly distinct light-inducible activities. While paCas13-1 demonstrated lower background activity, making it a potentially safer choice for certain applications, paCas13-2 demonstrated better RNA

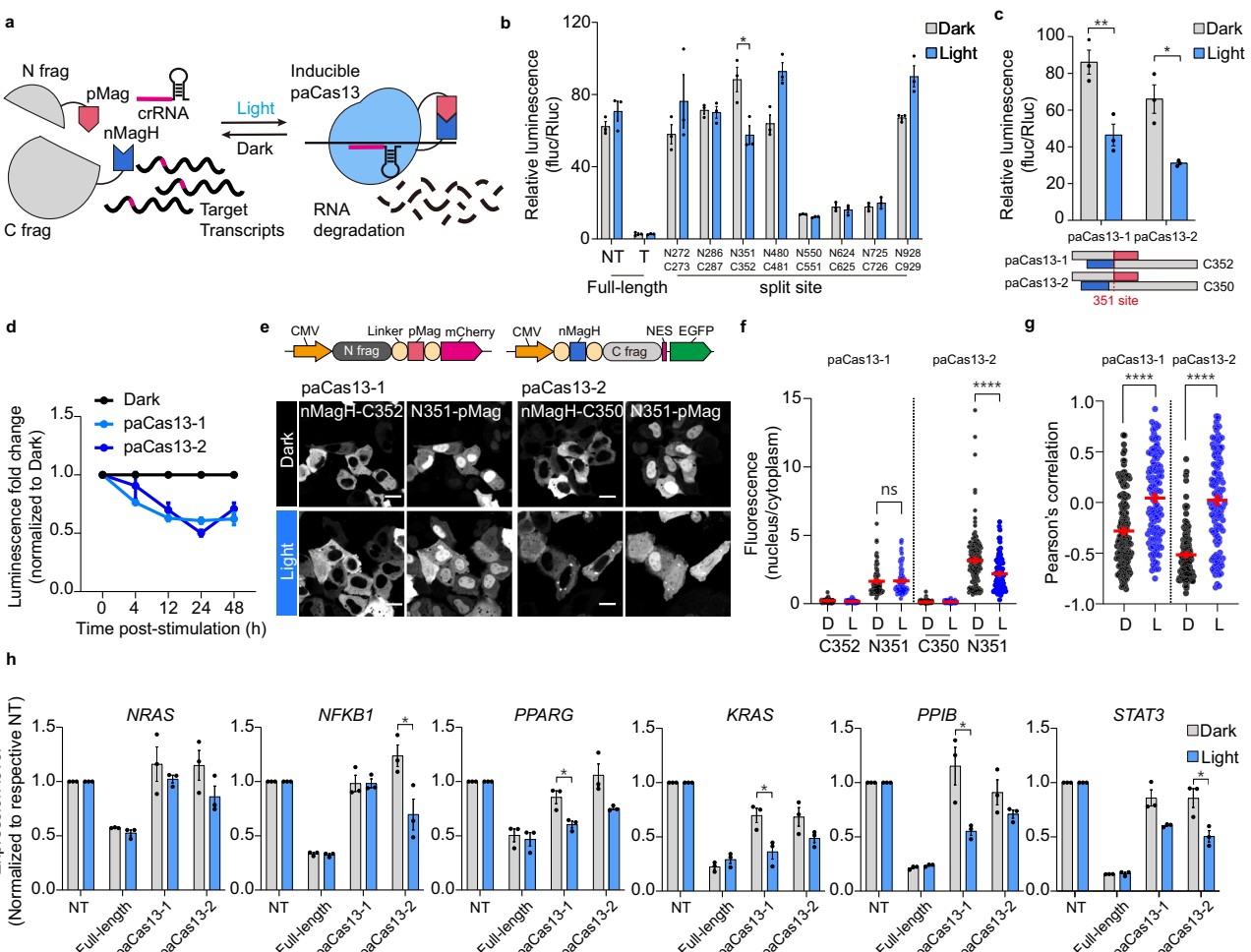

**Fig. 2 | Design and characterization of paCas13. a** Schematic diagram of the photoactivatable Cas13 (paCas13). Cas13 is split into two fragments, each fused to photoinducible dimerization domains. When exposed to blue light, pMag and nMagH heterodimerize, enabling the split Cas13 fragments to reassemble and undertake crRNA-mediated RNA degradation. **b** Light-induced disruption of luciferase activity using paCas13 fragments coupled to photoinducible dimerization domains in the absence or presence of light (1-min pulses delivered at 5-min intervals for 24 hours with an LED plate). **c** Schematic design of paCas13 candidates based on split-region configuration and comparison of light-inducible activity between paCas13-1 and paCas13-2. **d** Time-course analysis of paCas13 activity in the presence of light, as determined by the luminescence fold change normalized to that of each dark group (In the 4-hour groups of paCas13-2, $n = 3$ and in the other groups, $n = 5$ independent experiments). **e** Construction of paCas13 C-terminally fused with fluorescence protein and representative images of Hela cells co-

transfected with this paCas13. Scale bar = 20 μm. **f** The nucleus-to-cytosol ratio for each fragment of the paCas13 system, generated using data derived from the cells shown in panel (**e**). Dark of paCas13-1, $n = 86$ cells; Light of paCas13-1, $n = 90$ cells; Dark of paCas13-2, $n = 162$ cells; Light of paCas13-2, $n = 99$ cells. **g** Pearson's correlation analysis assessing the co-localization between the fluorescence signals of the N-terminal and C-terminal paCas13 fragments within the subcellular compartments of Hela cells, generated using data derived from the cells shown in panel (**e**). Dark of paCas13-1, $n = 157$ cells; Light of paCas13-1, $n = 124$ cells; Dark of paCas13-2, $n = 138$ cells; Light of paCas13-2, $n = 127$ cells. **h** Light-induced RNA knockdown of endogenous transcripts in HEK 293 T cells transfected with full-length or paCas13 candidates. All values were normalized to the corresponding non-targeted (NT) crRNA group within each dark and light condition. For panel **b**, **c**, and **h**, $n = 3$ independent experiments. *$P < 0.05$, **$P < 0.01$, ***$P < 0.001$, ****$P < 0.0001$ as determined by two-tailed Student's t-test. All error bars represent the mean ± s.e.m.

targeting efficiency to achieve more rapid and stable RNA degradation.

As we sought to optimize the paCas13 system, we examined the alternative split Cas13 sites reported in recent studies[10]. To enable a more precise evaluation, we also targeted the λ2 sequence for luciferase readout; this provided an exogenous indicator and allowed us to monitor the system's activity more clearly under induced and non-induced states. We compared the performance of the N351/ C350 and N761/C762 split sites under distinct inducible conditions, and we found that the N351/C350 split site exhibited high inducibility with substantially lower background activity than the previously characterized N761/C762 split site (Supplementary Fig. 3).

Since full-length PspCas13b was constructed with a C-terminal nuclear export sequence (NES)[3], the cellular localization of each Cas13 fragment was investigated to confirm their stable expression. Different

fluorescent proteins were fused to the C-terminus of each paCas13 pair, and their co-expression was examined (Fig. 2e). In the same cell under dark conditions, the EGFP-fused C-fragment displayed robust cytoplasmic translocation, while the mCherry-fused N-fragment was expressed throughout the entire cell, but displayed a predominant nuclear expression.

To investigate the factors that could influence cellular distribution, we performed a comprehensive investigation across all split candidates (Supplementary Fig. 4a). To determine the subcellular distribution of the HEPN domain, which is responsible for RNA cleavage, we constructed a fusion protein consisting of the HEPN domain and the mCherry fluorescent protein, and then performed real-time cellular monitoring. The expression was generally uniform across transfected cells, but we observed cellular toxicity and aggregation at 48 hours post-transfection in cells expressing the HEPN1 domain

(Supplementary Fig. 4b). When split candidates were expressed individually, we observed nuclear localization patterns specifically for the only these N286, N351, and N624 fragments. The N286 fragment, the leading sequence to contain the positively charged inter-domain linker (IDL)[15], was also the initial fragment to display a nuclear pattern (Supplementary Fig. 4a, c). This suggests that the IDL could be involved in directing N351 to the nucleus. The N624 fragment, meanwhile, demonstrated the distinct behavior of localizing in the nucleus when expressed alone and in the cytosol when co-expressed with its C625 partner (Supplementary Fig. 4c, d). This observation aligns with the experimental and computational data, which indicated that the N624/C625 combination has high auto-assembly activity and wide interfaces (Supplementary Fig. 2c). Interestingly, other fragments with high auto-assembly activity tended to form aggregates, possibly due to instability (Supplementary Fig. 4c, d). These observations suggest that electrostatic properties and protein-protein interactions contribute to the subcellular localization of the paCas13 system, which could affect its inducibility and background activity.

Building on these insights into the subcellular localization of these fragments, we examined how light stimulation could further modulate their interactions and subsequent functionality. After 24 hours of light stimulation, we observed a significant increase in the correlation between the N-terminal and C-terminal fragments, suggesting that the cytoplasmic recruitment of N351 was enhanced (Fig. 2e–g). These results support our contention that the paCas13 system functions as intended and provides further insights into its underlying mechanism. Although the N-fragment showed predominantly nuclear expression, enough was translocated to the cytoplasm to enable them to interact in that compartment.

Next, to confirm the ability of the paCas13 system to apply RNA interference, we targeted endogenous transcripts that had been previously targeted by a CRISPR-Cas13 system[3,8]. Although the paCas13 system does not reach the nuclease activity level of full-length Cas13, we observed significant light-induced RNA interference compared to dark conditions using paCas13-1 and −2 (Fig. 2h and Supplementary Tables 1, 2). To validate the robustness of these results, we included additional controls, including a cell viability test and an assessment of how light affects endogenous transcripts (Supplementary Fig. 1c, d). These controls confirmed that no significant changes were attributable to light exposure alone. Together, our results indicate that the paCas13 system can perturb endogenous transcripts through light illumination, with different levels of efficiency seen for paCas13 systems utilizing different split sites.

## Optogenetically inducible RNA base editing by the padCas13 editor

Several techniques that use base editing composed of catalytically inactive Cas13 (dCas13) to deliver targeted RNA for therapeutic applications have been reported for certain diseases[3,20]. In these systems, dCas13 fused with the regulatory protein was generally expressed, such that the targeted RNAs were continuously affected with no spatiotemporal resolution.

To develop a light-inducible base editing system, we fused catalytically inactive paCas13 fragments with the ADAR2 (adenosine deaminase acting on RNA type 2) domain (ADAR2$_{DD}$), which deaminates adenosine to inosine (which pairs with cytosine) in duplex RNAs[3,21]; this generated a system that we called the padCas13 editor (Fig. 3a). To characterize the activity of the padCas13 editor, we fused the wild-type ADAR2$_{DD}$ to the C-terminal ends of the C-fragments and constructed an RNA-editing reporter from the *firefly* gene of the dual-luciferase plasmid by generating a nonsense mutation, W417X (UGG to UAG). Unlike the wild-type reporter, the designed RNA-editing reporter contained a premature stop codon and exhibited no luciferase signal (Fig. 3a and Supplementary Fig. 5a). This mutation could be restored to the wild-type codon through A-to-I (adenine-to-inosine) editing, which

would recover the firefly luciferase signal. To identify the optimal guide position and design for padCas13 editors, we also designed targeted crRNAs by tiling with 30- or 50-nucleotide spacers across the target region based on mismatch distance (Fig. 3b). The chemical and light-inducible editor systems both showed restoration of the luciferase signal by RNA editing effects at 24 hours post-induction in the tiling assay (Fig. 3c, Supplementary Fig. 5c, and Supplementary Table 3). The most effective tiled crRNAs in our system were those with a 50-nucleotide spacer and a 40-nucleotide mismatch distance. The padCas13-1 and padCas13-2 editors displayed similar editing patterns, with padCas13-2 exhibiting more efficient RNA editing (Fig. 3c). The crRNA-tiling-dependent editing patterns of the padCas13 editors were highly correlated with one another when compared to full-length Cas13 (Supplementary Fig. 5d, e). This was especially evident when the editing effects were normalized to a 0-1 scale; we obtained Pearson R values of 0.76 and 0.91 for padCas13-1 and −2, respectively.

To determine the efficacy and reversibility of RNA base editing with the padCas13 editor, we performed a time-course assay. The restoration of luciferase activity by padCas13 editors increased as the time of light stimulation increased without affecting cell proliferation and viability (Fig. 3d, g). To further validate the restoration of luciferase activity, we directly validated the level of RNA base editing by analyzing sequencing chromatograms generated for the targeted RNAs by reverse transcription-polymerase chain reaction (RT-PCR)[22]. We conducted time-course light stimulation experiments, measured RNA editing levels for up to 48 hours. The level of A-to-I RNA editing at the targeted region increased gradually over time, and the flat, dark-colored G (I) chromatogram signal increased in a time-dependent manner (Fig. 3e, f). Additionally, a comparative analysis under only dark conditions between non-targeted (NT) crRNA and targeted crRNA revealed a measurable increase in basal editing levels, indicating that additional factors, such as the crRNA itself, may contribute to the background editing activity (Supplementary Fig. 5b). Following these findings, we next investigated the reversibility of RNA base editing mediated by the padCas13 editor. When the light was turned off after 6 hours, luciferase activity returned to baseline within 24 hours (Fig. 3h). Further analysis of RNA editing levels also demonstrated reversibility, with editing levels aligning with the dark state within 24 hours (Fig. 3i). These results suggest that the padCas13 editor enables reversible RNA base editing under light with different kinetics observed between RNA and protein synthesis and translation.

Next, to improve the base editing effect of padCas13 editor, we replaced the ADAR2 deaminase domain with one harboring hyperactive mutations[23]. We observed that ADAR2$_{DD}$ with E488Q and T490A mutations exhibited an increased restoration rate in our light-inducible system compared to other ADAR2$_{DD}$ variants[3,23], but had a higher base-editing signal in the dark state. To investigate the background activity of base editing, we transfected only the ADAR2$_{DD}$ (E488Q and T490A)-fused C-fragment of the padCas13 editor. Our results showed that the increased background signal appeared to depend on the ADAR2 variant but was unrelated to the specific C-fragment of padCas13 (Supplementary Fig. 6a, b). To verify this, we generated an additional group that carried stdMCP-ADAR2$_{DD}$ (E488Q and T490A) but not Cas13. Our results confirmed that the luciferase activity correlated with the ADAR2$_{DD}$ (Supplementary Fig. 6c). Because double-stranded RNA is required for the function of base editing, we also transfected the non-targeted and targeted crRNA. We found that the overexpression of the targeted crRNA had slight effects on both the luciferase restoration activity and the RNA editing rate (Supplementary Fig. 6c). Therefore, ADAR2$_{DD}$ protein overexpression can affect RNA editing in the absence of a crRNA.

To further confirm the base editing capabilities of our padCas13 editor, we compared our selected split site, N351/C350, with the previously characterized N761/C762[10] under various induction conditions. This comparison demonstrated that our chosen site has lower

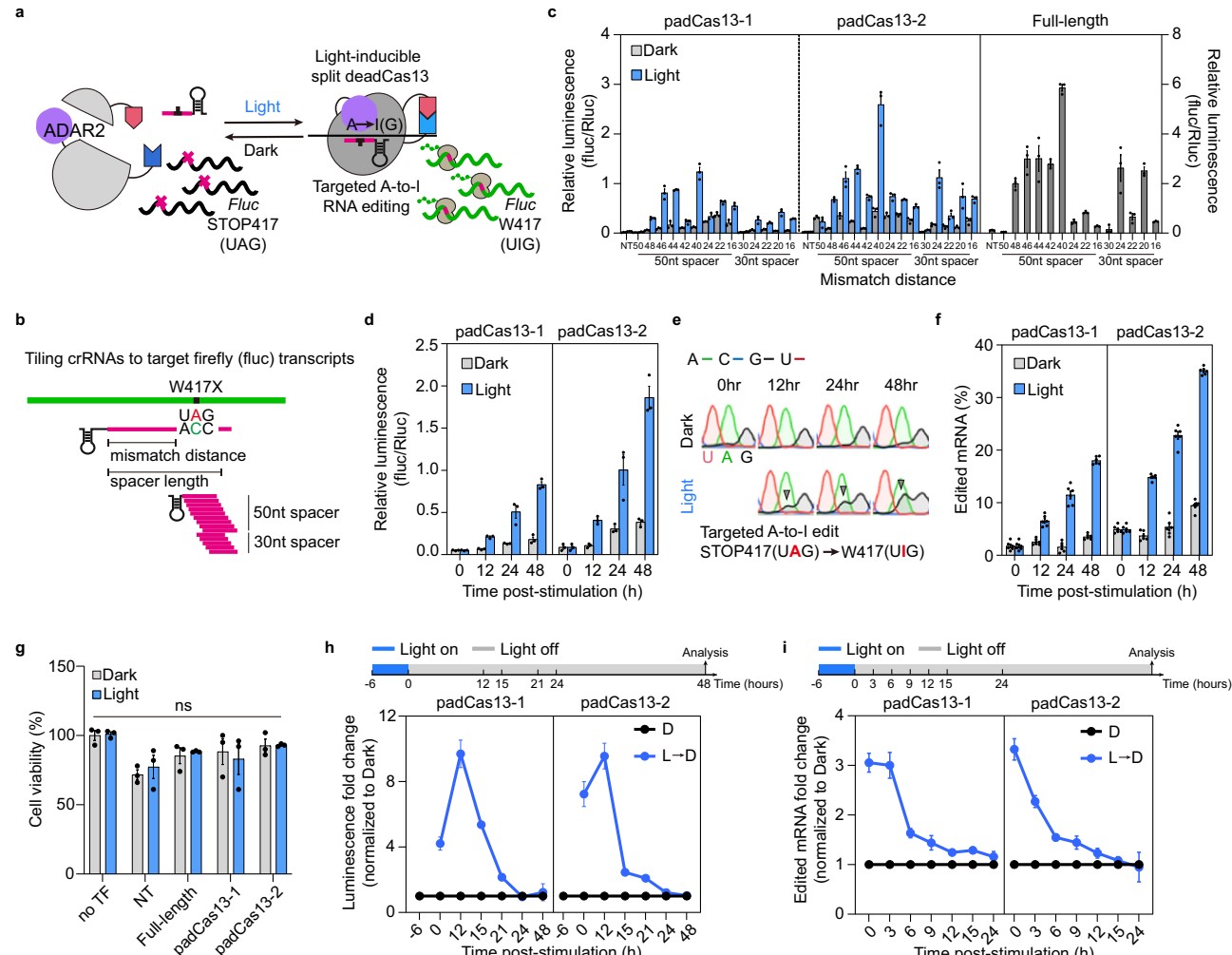

**Fig. 3 | Optogenetic A-to-I RNA editing and characterization of padCas13 editor.** **a** Schematic diagram of targeted RNA editing using padCas13 (containing the Cas13 HEPN RNase domain-inactivating mutations, H133A and H1058A) with ADAR2_DD and overview of the A-to-I RNA-editing reporter assay. A single G-to-A mutation was introduced into the *firefly* luciferase sequence; this caused the W417X mutation (TGG to TAG), which did not enable generation of a detectable firefly luciferase signal. **b** Tiling crRNAs designed to target W417X mutant firefly luciferase transcripts using spacers of two different lengths (50 and 30 nucleotides). Spacer length represents the region between the guide and target sequences. Mismatch distance was measured between the 3′ end of the spacer and the mismatched cytidine. **c** Restoration of light-induced luciferase activity via A-to-I RNA editing with tiled crRNAs in the presence or absence of light. Each full-length Cas13 and padCas13 editor was coupled with ADAR2_DD (E488Q/T490A). (*n* = 3 independent experiments) **d** Time-course analysis of padCas13 editor activity in the presence of light, as determined by restoration of the firefly luciferase signal (*n* = 3 independent experiments). **e** Sequencing chromatograms are used for editing quantification

under dark (top) or light (bottom) conditions. Gray triangles indicate increased G level. **f** Direct quantification of RNA editing percentages using RT-PCR-Sanger sequencing (*n* = 6 independent experiments). **g** Cell viability effects of transfected constructs related to inactive Cas13 under dark and light conditions, analyzed using two-way ANOVA (*n* = 3 independent experiments). **h** Reversibility of light-induced luciferase restoration by the padCas13 editor was assessed after blue-light illumination was removed. The data obtained at each time point were normalized to those obtained from the corresponding dark-exposed control group (*n* = 3 independent experiments). **i** Direct quantification of RNA editing percentages was performed using RT-PCR-Sanger sequencing after blue-light illumination was removed. The data obtained at each time point were normalized to those obtained from the corresponding dark-exposed control group (*n* = 3 independent experiments). For **c**, **d**–**f**, and **g**, blue light was delivered with 1-min light pulses delivered at 5-min intervals for 24 hours with an LED plate. All error bars represent the mean ± s.e.m.

background activity and is more suitable for RNA base editing (Supplementary Fig. 7). We also examined more cell types for the potential application of padCas13 editor. We observed light-inducible RNA base editing in human cervical carcinoma HeLa, breast cancer MCF7, fibrosarcoma HT1080, and mouse neuroblasts Neuro-2a cells (Supplementary Fig. 8). Across the alternative split site and different cell lines examined, the padCas13 editor demonstrated consistent light-inducible RNA base editing capabilities, underscoring its versatility and broad potential for diverse therapeutic applications.

To expand the range of disease mutations and protein modifications that can be targeted, we developed a light-inducible C-to-U (cytidine-to-uridine) RNA editing system by fusing padCas13 fragments

to an evolved ADAR2_DD capable of cytidine deamination[20]. To validate the activity of the padCas13 editor for C-to-U base editing, we designed a fluorescent protein-based indicator by introducing the Y66H green-to-blue mutation, Y66H (UAC to CAC), into GFP and developed a crRNA that would directly target the CAC codon with a 50-nucleotide spacer that included a 34-nucleotide mismatch distance and a U flip, resulting in a C-to-U conversion (Fig. 4a, b). Under light conditions, the padCas13 editor would correct the Y66H mutation to generate GFP fluorescence, which could be measured by cell imaging and flow cytometry. We transfected this padCas13 editor with cytidine deaminase and a targeted crRNA harboring the GFP indicator, and then evaluated the light inducibility after 24 hours of blue-light

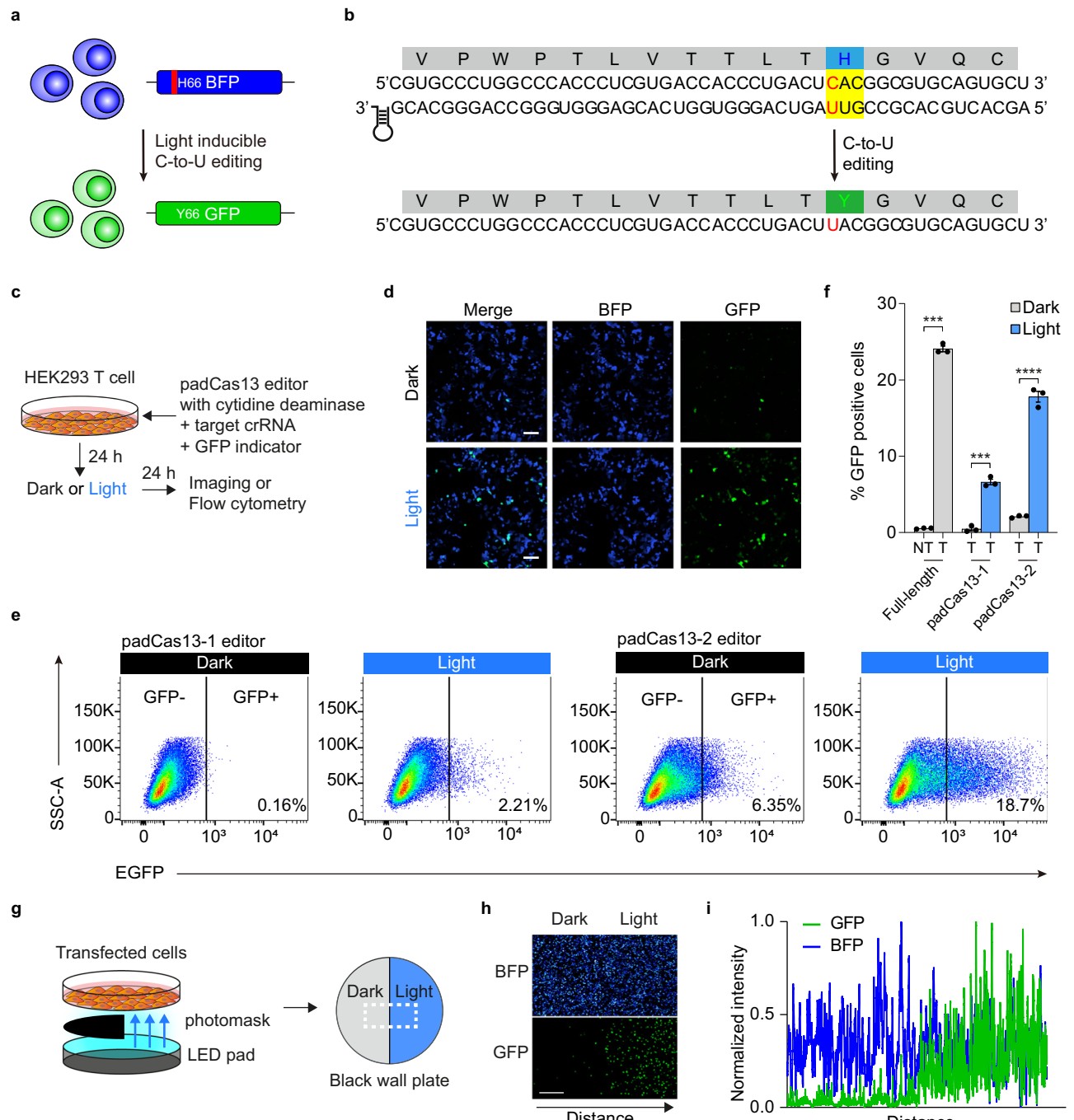

**Fig. 4 | Optogenetic C-to-U RNA editing by the padCas13 editor. a** Schematic of GFP reporter for light-induced C-to-U RNA editing activity. **b** EGFP mutant with H66Y target site and targeting designed guide RNA with a uridine mismatched base. **c** Schematic diagram depicting the restoration timeline of GFP H66Y mutation via padCas13 editor. Blue light was delivered at 5-min intervals for 24 hours with an LED plate. **d** Representative 40X confocal microscopy images of HEK 293 T cells co-transfected with the padCas13-2 editor, the GFP reporter, and the target crRNA. Scale bar = 100 μm. **e** Representative flow cytometry plots of cells transfected with the padCas13 editors with cytidine deaminase. FITC-A (log scale) is shown on the X-axis; SSC-A (linear scale) is shown on the Y-axis. **f** Quantification of GFP H66Y restoration by the padCas13 editor and full-length Cas13 for C-to-U RNA editing ($n$ = 3 independent experiments) ***$P$ < 0.001 and ****$P$ < 0.0001 as determined by two-tailed Student's t-test. **g** Schematic illustration of strategy for local activation using light-induced C-to-U RNA editing activity and a photomask. **h** Confocal microscopy 20X images of HEK 293 T cells after 24 hours of local light stimulation. Represented data from at least three independent experiments. Scale bar = 1 mm. **i** Quantitative analysis of fluorescence intensity in the imaged area of panel (**h**), as performed using ImageJ. Error bars represent the mean ± s.e.m.

stimulation (Fig. 4c). Confocal microscopy revealed that the padCas13 editor-induced C-to-U RNA editing, thereby generating GFP-positive cells under the light condition (Fig. 4d and Supplementary Fig. 9). Moreover, flow cytometry-based quantification demonstrated that the light-induced C-to-U editing of the padCas13 editor increased the population of cells positive for the GFP signal from 6% (dark condition)

to 18% (under light exposure) (Fig. 4e, f, and Supplementary Fig. 10). Additionally, we employed a photomask to demonstrate that we could spatially control padCas13 editor-mediated RNA editing (Fig. 4g–i). Strong GFP fluorescence was observed only on the light-exposed side after 24 hours of light stimulation. These results suggest that the padCas13 editor system enables light-induced, spatiotemporal

RNA base editing and can be expanded by replacing the regulatory protein.

## padCas13 editor enables endogenous RNA base editing

To determine whether our padCas13 editor system could be used to edit endogenous transcripts in mammalian cells, we examined previously identified disease-relevant target transcripts under light conditions, and measured the editing results via analysis of sequencing chromatograms[3,7,8,20,22]. The padCas13 editing system and targeted crRNA were transfected to HEK 293 T cells, and blue-light illumination was carried out for 24 hours. The results revealed that the padCas13 editors could mediate A-to-I and C-to-U editing of all evaluated endogenous targets with significant editing levels (Fig. 5a, b and Supplementary Tables 5, 6). Furthermore, we compared the editing efficiency

of the padCas13 editors and full-length Cas13 (Fig. 5a, b). While the padCas13 system exhibited slightly lower editing efficiency, the RNA editing activity obtained under light conditions remained significantly higher than that seen in the dark group and was within a range similar to that achieved with full-length Cas13.

Next, to demonstrate the ability of the padCas13 editor to edit the phenotypic function of specific targets, we used our system to adjust a post-translational modification (PTM) in the Wnt/β-catenin pathway. More specifically, we performed light-induced base editing of the phosphorylation status of key residues of β-catenin whose phosphorylated forms inhibit protein degradation[24]. To design our PTM reporter system, we integrated a TOPFlash system based on *firefly* luciferase genes[20,25] into a dual-luciferase assay system that enabled monitoring of signal induction. Under light conditions, the PTM

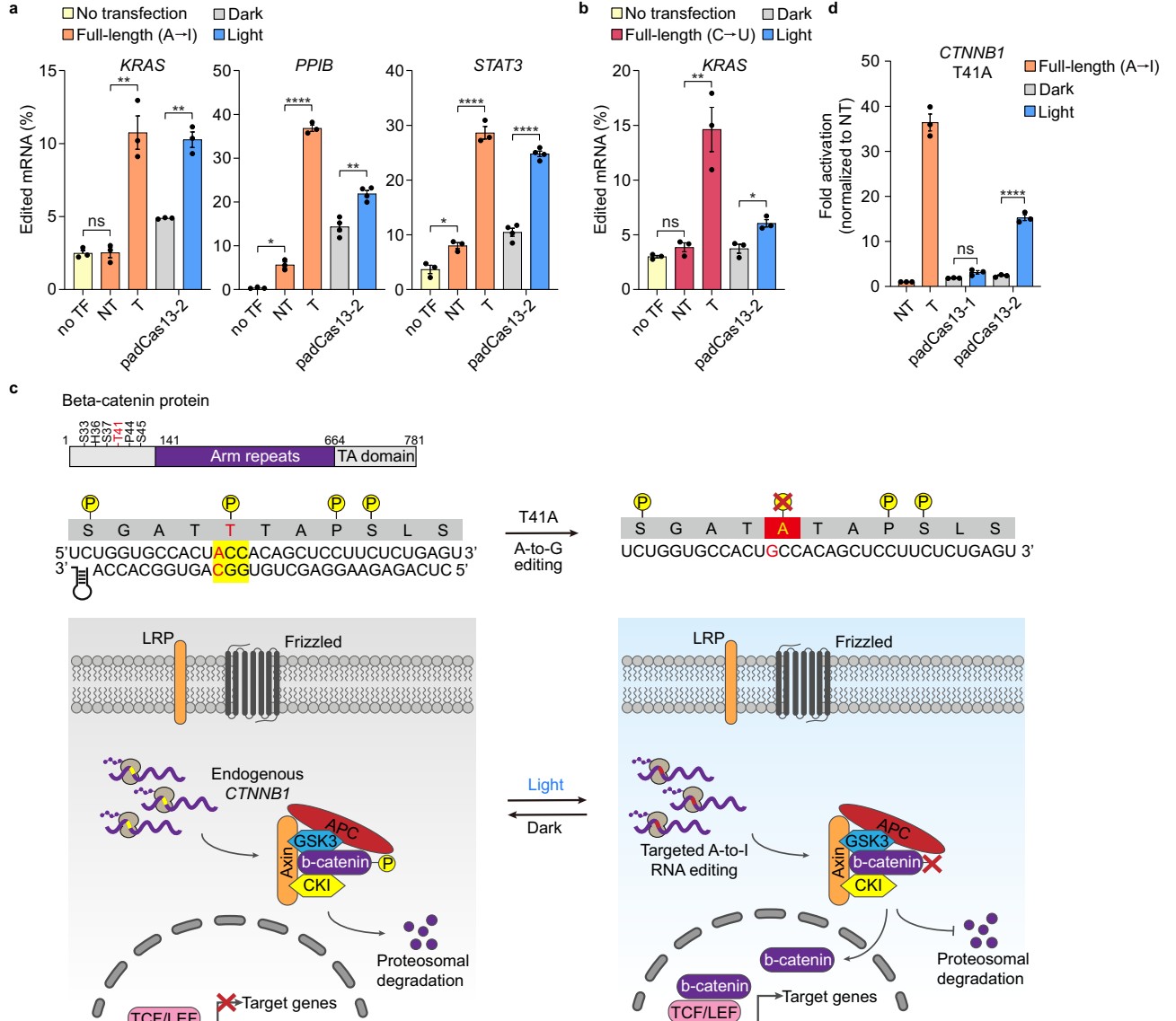

**Fig. 5 | Inducible editing of endogenous transcripts by the padCas13 editor.**
**a** RT-PCR-Sanger sequencing-based quantification of A-to-I RNA editing by the padCas13-2 editor with E488Q/T490A mutation of ADAR2$_{DD}$ after 24 hours of light stimulation. Cells were illuminated with blue light for 1-min at 5-min intervals for 24 hours with an LED plate. Groups are no transfection (no TF) and, for full-length Cas13b, non-targeted (NT) and targeted (T) crRNAs (*n* = 3 for groups of no TF and Full-length, and *n* = 5 for groups of padCas13-2 independent experiments). **b** RT-PCR-Sanger sequencing-quantification of C-to-U RNA editing by the padCas13-2

editor after 24 hours of light stimulation. Groups are as described in panel (**a**) (*n* = 3 independent experiments). **c** Schematic representation of Wnt/β-catenin signaling by RNA-edited *CTNNB1* by the padCas13 editor. **d** Activation of Wnt/β-catenin pathway by β-catenin/TCF/LEF complex via padCas13 editors in the presence of light. For full-length Cas13b, groups are NT and T crRNAs (*n* = 3 independent experiments). *P < 0.05, **P < 0.01, ***P < 0.001,****P < 0.0001 as determined by two-tailed Student's t-test. Error bars represent the mean ± s.e.m.

reporter will be induced by padCas13 editor-programmed alteration of phosphorylation among residues of β-catenin (Fig. 5c). We evaluated both padCas13-1 and padCas13-2 editors, as a means to assess how the editing efficiency affected the phenotypic change at the protein level. Similar to our luciferase reporter results, we found that padCas13-2 editor was more effective than padCas13-1 editor. We further confirmed light-inducible RNA editing by observing PTM reporter activation, which reflected Wnt/β-catenin pathway activation downstream of effective targeting of β-catenin transcripts (*CTNNB1*) (Fig. 5d). These results demonstrate that the use of the padCas13 editor can be expanded to target PTMs of target proteins, such as those relevant to pathological conditions.

## In vivo editing using the padCas13 editor

Several groups have recently reported in vivo genome editing with the CRISPR-Cas system[26–28]. This is important because the successful in vivo application of a system critically supports its potential for use in disease models. However, the vast majority of studies using CRISPR-based gene expression systems have been limited to the in vitro level, and no previous study has reported the use of a light-inducible CRISPR-Cas13 system in vivo. The latest developments in red-light inducible systems have expanded their in vivo applications[29,30], but such work is still in its early stages. Moreover, little has been done using blue-light activation systems, which are commonly applied in optogenetics.

To determine whether our padCas13 editor could function in an animal model, we delivered padCas13 editor constructs to mice via hydrodynamic tail-vein injection with a luciferase-based RNA editing reporter and targeted crRNA plasmids. We then shaved the abdomen fur from the mice and randomly classified them into dark and light groups. An LED plate was positioned directly under the mouse cage for illumination purposes (Fig. 6a). Due to the anatomical arrangement of the mouse, where the liver is located closer to the skin surface than in a

human, blue light can effectively penetrate the thin mouse skin and reach the liver tissues (Supplementary Fig. 11). Indeed, previous research demonstrated that blue light from an LED array can effectively reach mouse liver tissues[31]. Compared to mice injected with the padCas13-2 editor but not exposed to the light stimulation, the RNA editing-enabled mice exhibited a substantially greater luciferase signal (Fig. 6b–d and Supplementary Fig. 12a). Consistent with these findings, the padCas13 editor system successfully enabled RNA editing in vivo. The previous report of a Cas13-based RNA editing system (named REPAIR[3]) did not include information on its in vivo use. Here, we tested whether our padCas13 system was capable of RNA editing in vivo and observed the editing efficiency of the padCas13 editor was comparable to that of a full-length Cas13 system (Supplementary Fig. 12b). Moreover, to validate the efficacy of the padCas13 editor, we exposed transfected mice to blue light under various intensities (Dark, 0.25, 0.5, 1, and 5 mW/cm²). Our results indicated that the activity of the padCas13 editor was dependent on the intensity of illumination, with 1 mW/cm² proving to be sufficient for inducing RNA base editing in vivo (Supplementary Fig. 13a, b). We also observed the intensity-dependent luminescence signal in the livers isolated from these mice, confirming the in vivo applicability of the padCas13 editor under blue light conditions (Supplementary Fig. 13c, d). These findings demonstrate that the padCas13 editor facilitates in vivo RNA base editing when activated by blue light. To further validate the robustness and safety of the padCas13 editor, we evaluated potential liver damage by measuring levels of alanine aminotransferase (ALT), aspartate aminotransferase (AST), alkaline phosphatase (ALP), and total bilirubin. Biochemical analysis revealed that all of the measured parameters were within the clinically normal range, as defined by Charles River guidelines, for both the light-exposed and dark control groups (Supplementary Fig. 14a). Additionally, histological analysis via hematoxylin and eosin (H&E) staining of mouse liver tissues showed no

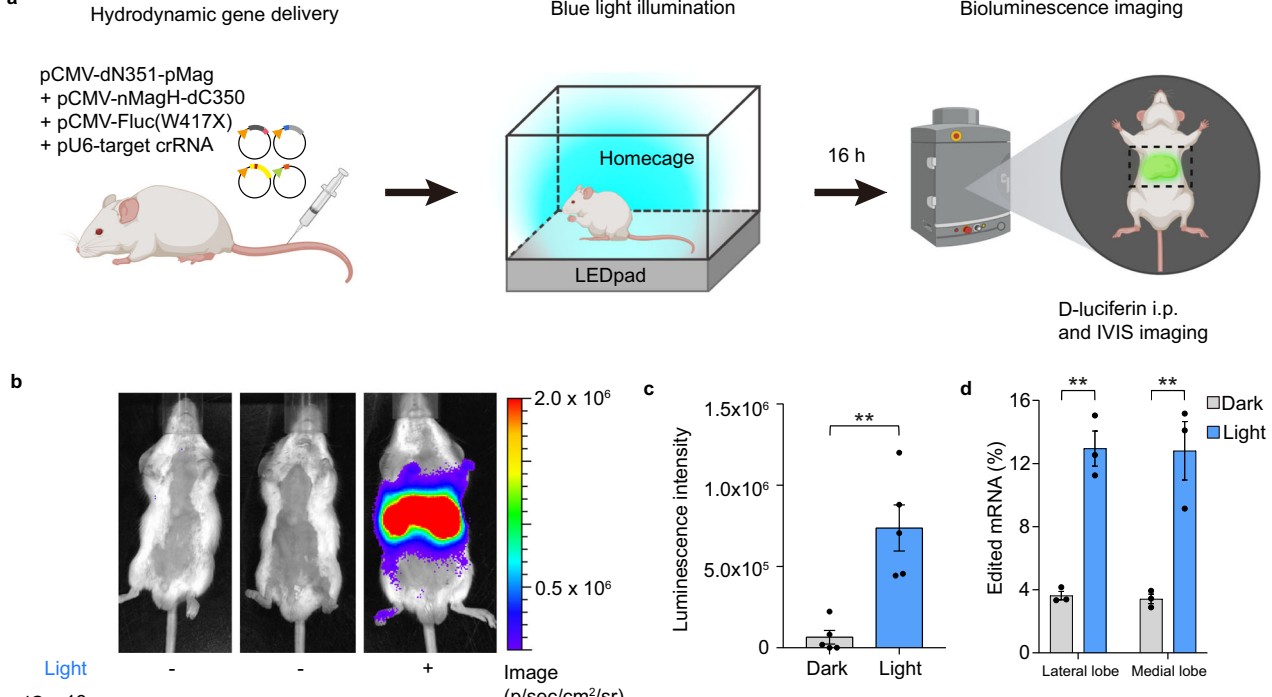

**Fig. 6 | In vivo RNA editing with padCas13 editor. a** Schematic representation of the procedure for RNA editing by the padCas13 editor under light stimulation with LED pad. Post tail-vein injection, mice were illuminated with LED light in their home cage and bioluminescence imaging was performed after 16 hours. **b** Luminescence images of mice with the luciferase reporter (W417X) and padCas13 editor.

**c** Quantification of total bioluminescence intensity for each mouse (n = 5 mice per group). **d** RT-PCR-Sanger sequencing-based quantification of A-to-I RNA editing by the padCas13-2 editor in the livers isolated from mice after 16 hours of light stimulation (n = 3 mice per group). **P < 0.01 as determined by two-tailed Student's t-test. Error bars represent the mean ± s.e.m.

observable differences between the light-exposed and dark control groups (Supplementary Fig. 14b). Our results suggest that the pad-Cas13 editor system can facilitate the robust activation of target transcripts in a mouse model.

## Discussion

Here, we developed a light-inducible RNA-modulating technique based on CRISPR-Cas13. The paCas13 and padCas13 editor systems can enable users to perform diverse manipulations of target RNAs in vitro and in vivo, including RNA degradation and base editing.

The developed paCas13 system presents distinct advantages over previous RNA editing technologies. Our utilization of the original PspCas13b crRNA avoids the need for additional engineering steps and provides a simpler approach for RNA base editing than methods involving caged guide RNAs[32,33] or split effector domains[34]. Regarding in vivo applications, the auto-assembly site sets of paCas13 can flexibly integrate with or without intein systems, expanding the utility of our system for therapeutic applications.

We also found that the auto-assembly activities of split Cas13 were similar to those of previously reported split Cas proteins[19,35]. Although several engineered split proteins have been reported, deciphering the mechanism of reconstitution is a challenge. Initial cell-based experiments demonstrated that there were distinct differences in the propensity for spontaneous reconstitution across the examined split sites, such as between N351/C352 and N550/C551 (Fig. 1f). To interpret these findings, we employed MaSIF-site based computational analysis. This analysis revealed that a higher degree of interface propensity was associated with increased auto-assembly activity (Supplementary Fig. 2).

Recent research on CRISPR-Cas13 systems has focused mainly on using the compact Cas13 protein for in vivo delivery[7,8]. CRISPR-based strategies typically involve fusing a regulatory protein to the Cas protein; this generates fusion genes that are too large to be packaged in an AAV vector, limiting the utility of these compact systems[36]. Our paCas13 components are shorter than full-length Cas13 and can be packed into separate AAV vectors, increasing their relevance for in vivo RNA modulation. The use of auto-assembled split proteins also has the potential to be implemented in vivo as an RNA degradation approach.

Although our paCas13 system performs efficient and spatiotemporally regulated RNA base editing, further optimization is needed to increase its efficiency to that of the full-length Cas13b system. The requirement for additional activation inducers, such as rapamycin or light exposure, may constrain the overall activity of the splitting-based system. RNA targeting also presents inherent complexities, especially compared to DNA targeting systems, such as CRISPR-Cas9. While edited DNA remains permanently altered, targeting of RNA is inherently transient due to the continuous nature of the transcription, splicing, translation, and degradation processes. The efficiency and robustness of the light-inducible system must be enhanced further. The paCas13 system should also be adapted for AAV delivery to broaden its application scope. At this stage, the paCas13 system reported herein should be viewed as an initial proof-of-concept that demonstrates the potential for light-inducible control.

To address the above-listed limitations, future research could explore the use of directed evolution to enhance system efficiency, enable high-throughput screening, and facilitate efficient exploration for reducing background noise. The ability of directed evolution to efficiently optimize enzyme activity has been evidenced in other systems, such as those involving compact prime editors[37,38] and evolved ADAR[20], the latter of which was also utilized in this study. Moreover, given the recent advancements in utilizing optogenetic tools for directed evolution[39], it is feasible that paCas13 could be adapted to employ this strategy. Direct evolution under specific light conditions may offer a way to develop a more efficient RNA knockdown or editing

tool with minimal background effects. Such an approach could also be flexibly adapted to various other inducible systems.

In sum, we herein demonstrated systems for light-inducible RNA interference, A-to-I editing, and C-to-U editing. By switching the utilized regulatory proteins, users could further apply paCas13 and padCas13 editor for RNA labeling, splicing, epitranscriptomic modification, and post-translational modification. We utilized the PspCas13 protein to create paCas13 and padCas13 editor via chemical-based split site screening. We anticipate that in vivo approaches using the paCas13 system will be broadly applicable to the manipulation of RNA in a variety of disease states and physiological processes.

## Methods

### Ethical statement

All animal experiments and procedures complied with the Institutional Animal Care and Use Committees (IACUC) guidelines at the Korea Advanced Institute of Science and Technology (KAIST). The approval number for this study is KA2023-089-v1. Six-week-old male BALB/c were purchased from Raonbio Service (Yongin-si, South Korea).

### Construction of inducible Cas13

N- and C-terminal fragments of codon-optimized *Prevotella sp. P5-125* Cas13b were amplified from pC0046-EF1a-PspCas13b-NES-HIV (Addgene plasmid 103862). In this study, the N/C double numbering scheme (e.g., N351/C352) is used to specify the exact boundaries of the split sites, allowing for the clear indication of both adjacent and non-adjacent amino acids. To construct padCas13, inactivated HEPN RNase domains (containing H133A and H1058A mutations) were amplified from pC0049-EF1a-dPspCas13b-NES-HIV (Addgene 103865). To avoid steric hindrance between split Cas13 fragments and inducible domains (FKBP-FRB or pMag-nMagHigh1), a 16 amino acid-long Gly-Ser linker was cloned between each Cas13 fragment and the inducible domain[19]. Inducible domain-encoding DNA sequences were synthesized by Twist Bioscience. All inducible Cas13 constructs were generated using Gibson assembly. The PCR fragments used for Gibson Assembly were amplified using Phusion Hot-Start II DNA polymerase (ThermoFisher). The inducible domain-fused Cas13 constructs of each pair were cloned, respectively, into the XhoI/XmaI and BsrGI/NotI sites of the EGFP-N1 plasmid (Takara).

For the split-fluc reassembly assay, the N-fragment of Cas13 was fused with the C-fragment of firefly luciferase and the C-fragment of Cas13 was fused with the N-fragment of firefly luciferase[17]. All constructs were generated by Gibson assembly; fragments from the pmirGLO dual-luciferase reporter vector (Promega) and pC0046-EF1a-PspCas13b-NES-HIV (Addgene plasmid 103862) were inserted into the BamHI and NotI sites of the EGFP-N1 plasmid (Takara).

For live-cell imaging of paCas13 system activity under dark and light conditions, N351-pMag was fused with the mCherry protein, and nMagHigh1-C352 (for paCas13-1) or nMagHigh1-C350 (for paCas13-2) were linked with EGFP. The constructs were generated using Gibson assembly, and DNA fragments corresponding to N351-pMag, nMagHigh1-C352, nMagHigh1-C350, EGFP-N1 (Takara), and mCherry-C1 (Takara) were inserted into the BamHI and NotI sites of the EGFP-N1 plasmid (Takara).

### Construction of crRNAs

The crRNAs targeting the Fluc reporter, the GFP indicator, *NRAS, NFKB1, PPARG, KRAS*, *PPIB*, *STAT3*, and the A-to-I or C-to-U RNA editing sites were generated by annealed oligo cloning using the BbsI site of the pC0043-PspCas13b crRNA backbone (Addgene plasmid 103854). All crRNA plasmids were cloned using T4 DNA ligase (Elpis Biotech).

### Reporter constructs for RNA editing

The reporter for screening the split site was generated by annealing oligos corresponding to complementary λ2 bacteriophage target

sequences[16] and inserting them between the NheI and XhoI sites of the pmirGLO dual-luciferase reporter vector (Promega) to generate the dual-luciferase-λ2 target site.

The ADAR2 deaminase domain (ADAR$_{DD}$) for A-to-I editing was amplified from pC0050-CMV-dPspCas13b-longlinker-ADAR2$_{DD}$ (wt) (Addgene plasmid 103866).

For hyperactivated A-to-I RNA editing, we used the ADAR2$_{DD}$ (E488Q/T490A) mutant[23], which was generated by point mutation and Gibson assembly from pC0050-CMV-dPspCas13b-longlinker-ADAR2$_{DD}$ (wt).

For C-to-U RNA editing, the evolved ADAR2$_{DD}$ (containing E488Q/V351G/S486A/T375A/S370C/P462A/N597I/L332I/I398V/K350I/M383L/D619G/S582T/V440I/S495N/K418E/S661T mutations) was amplified from pC0079 RESCUE-S (Addgene plasmid 130662).

The reporter vector was generated by introducing the W417X mutation (TGG to TAG) into the pmirGLO dual-luciferase reporter vector (Promega) through Gibson assembly of two fragments from the firefly luciferase gene of the pmirGLO dual-luciferase reporter vector (Promega). This reporter vector expressed Renilla luciferase as a normalization control but had impaired firefly luciferase expression.

For C-to-U RNA editing experiments, the Y66H green-to-blue mutation was introduced into the EGFP-N1 plasmid by Gibson assembly. To increase the effect of C-to-U RNA editing, we cloned two vectors by extension PCR cloning followed by Gibson assembly. The vector expressed the crRNA and a pMag-conjugated N-terminal dCas13 fragment linked with the GFP indicator via a T2A self-cleaving peptide. These sequences were driven by the hU6 promoter and EF1a short (EFS) promoter, respectively. The second vector expressed the C-terminal fragment of dCas13 fused with nMagHigh1 and evolved ADAR2$_{DD}$ under the control of the EFS promoter.

To visualize *CTNNB1* base editing by a phenotypic change, a luciferase reporter vector was generated from M50 Super 8x TOPFlash (Addgene plasmid 12456)[25]. The reporter vector was generated by Gibson assembly; fragments containing TCF binding sites and minimal promoter from the M50 Super 8x TOPFlash and *firefly* luciferase gene were inserted into the BglII and BsrGI sites of the pmirGLO dual-luciferase reporter vector (Promega).

## Prediction of protein structure and molecular surfaces

Protein sequences of full-length PspCas13b, the C-fragment of PspCas13b, and the N-fragment of PspCas13b were each used as query sequences for protein structure predictions through ColabFold (AlphaFold2 with MMseqs2) (Open-source software at https://github.com/sokrypton/ColabFold)[13,14]. The sequence of chain A was extracted, and multiple sequence alignments were constructed using the automated MMseqs2 in ColabFold, employing three sequence recycles, which control the number of times the prediction is reprocessed through the model (default is 3). The pLDDT score indicates the confidence level for how well the predicted structure is expected to align with an experimentally resolved structure. A higher pLDDT score signifies increased confidence in the protein structure prediction. pLDDT > 90 (Highly Accurate), 70-90 (Well-Modeled), 50-70 (Low Confidence).

AlphaFold2 extracted PDB files of input sequences, which were then processed by the MaSIF-site pipeline (scripts at https://github.com/lpdi-epfl/masif) to predict protein-protein interactions[18]. The analyzed protein files were visualized using PyMOL.

For quantification of interface properties across split sites, the total interface ratio is calculated by taking the number of atoms in N-fragment and C-fragment with a score of 0.5 or higher, divided by the sum of the total number of atoms in N-fragment and C-fragment.

## Cell culture and transfection

HEK 293 T (CRL-11268), HeLa (CCL-2), MCF7 (HTB-22), HT1080 (CCL-121), and Neuro-2a (CCL-131) cells were purchased from American Type Culture Collection (ATCC). HEK 293 T, HeLa, and Neuro-2a cells were cultured in Dulbecco's modified Eagle's medium (Gibco) supplemented with 10% fetal bovine serum (FBS) (Corning) at 37°C with 10% or 5% $CO_2$ under standard humidity. MCF7 and HT1080 cells were cultured in Eagle's minimum essential medium with 10% FBS (Corning) at 37°C with 10% $CO_2$ under standard humidity. The cells were transfected using Lipofectamine LTX (Invitrogen) according to the manufacturer's recommendations.

## Luciferase reporter assay

To measure the efficiency of RNA knockdown, HEK 293 T cells were co-transfected with 10 ng of dual luciferase reporter plasmid carrying the λ2 sequence[40], 75 ng of each inducible Cas13 fragment-expressing plasmid, and 100 ng of crRNA expression plasmid. Prior to transfection, the cells were seeded on 96-well plates (Corning) and allowed to grow until they reached 80% confluence (~16 hours). The next day, 10 μL Opti-MEM I reduced serum medium (ThermoFisher) containing 0.6 μL Lipofectamine LTX was combined with 10 μL Opti-MEM I reduced serum medium containing 0.26 μL PLUS reagent and all relevant DNA plasmids. The solutions were mixed, incubated for 5 min, and then slowly added to each well. After 24 hours, the transfected cells were induced with light illumination using an LED plate (488 nm, 100 μW/cm²) with 1 min pulse every 5 min for 24 hours. The firefly and Renilla luciferase readouts were monitored using the Dual-Glo Luciferase Assay System (Promega) and the luminescence scan module of a Tecan microplate reader according to the manufacturer's instructions. The firefly luciferase signal was normalized to the corresponding Renilla signal, and the normalized difference in the protein level of firefly luciferase was calculated.

For the split firefly luciferase assay, HEK 293 T cells were co-transfected using the same protocol as described above, with the only differences being 125 ng of each split firefly fused -Cas13 fragment expressing plasmid and 10 ng of Renilla luciferase reporter plasmid as the internal control.

## Cell viability assay

Cell viability was assayed using the CellTiter 96 Aqueous one solution cell proliferation assay (Promega) according to the manufacturer's instructions. HEK 293 T cells were transfected with the paCas13 or padCas13 system with crRNAs and illuminated with blue light using an LED plate (488 nm, 100 μW/cm²) with 1 min pulse every 5 min for 24 hours. Then, 20 μl of the solution was put into each well of the 96-well plate in 100 μL of culture medium. The samples were mixed gently and incubated at 37 °C for 1 hour in a humidified, 10% $CO_2$ and the absorbance at 490 nm using a Tecan microplate reader according to the manufacturer's instructions.

## Immunofluorescence

HEK 293 T cells expressing each paCas13 fragment were fixed with 4% paraformaldehyde at room temperature for 10 minutes and washed with PBS three times. Following cell permeabilization with PBS containing 0.1% Triton X-100 (Sigma) for 10 min and washed three times, the cells were blocked by incubation for 1 hour in PBS containing 1% BSA and incubated overnight at 4 °C with the primary antibodies rabbit monoclonal anti-HA (Cell Signaling Technology, 3724;1:1000) and rabbit monoclonal anti-FLAG M2 (Cell Signaling Technology, 14793;1:1000) diluted in PBST containing 1% BSA. The cells were washed in PBST and incubated at room temperature for 1 hour with Alexa Fluor 488-conjugated goat anti-rabbit secondary antibody (Invitrogen, A-11034;1:1000). After washing with PBST, the cells were imaged by confocal microscopy.

## Endogenous RNA knockdown assays

To measure RNA levels after transfection of the paCas13 system, HEK 293 T cells were plated on 48-well plates (Corning) and transfected

with 250 ng of each padCas13 fragment expressing plasmid and 300 ng of crRNA plasmid. For a positive control, cells were transfected with 500 ng of full-length Cas13 expressing plasmid and 300 ng of crRNA plasmid. For a negative control, 800 ng of firefly luciferase reporter plasmid was transfected as a vehicle. After 24 hours, the paCas13 system was illuminated by blue light for 24 hours. Total RNA was isolated using a PureLink RNA mini kit (Invitrogen) and reverse transcribed to cDNA using a PrimeScript RT Reagent Kit with gDNA eraser (Takara). All qPCR reactions were run in 20-µL volumes using SYBR green master mix (Toyobo) and a CFXmaestro 96 instrument (Bio-Rad). qPCR primers were designed to flank the crRNA-targeted site. Expression levels were calculated using the cycle threshold ($C_t$) values of a housekeeping control gene (GAPDH) and the targeted gene. The relative expression level was determined by the equation $2^{\Delta Ct}$, where $\Delta C_t = C_t$ (sample) - $C_t$ (control). The relative target gene expression level was obtained by normalizing the target gene expression level in on-target crRNA-treated cells relative to that in non-targeted (NT) crRNA-treated cells.

## RNA editing activity quantification

To quantify the editing activity of the padCas13 editor system, HEK 293 T cells were plated on 48-well plates (Corning) and transfected as described above. Total RNA was harvested and purified using a Pure-Link RNA mini kit (Invitrogen), and reverse transcribed by 2 µM of gene-specific primers or oligo dT and random hexamer primers taken from the PrimeScript RT Reagent Kit with gDNA eraser (Takara). Each cDNA product was PCR-amplified with Phusion DNA polymerase (ThermoFisher) using target-specific primers and sent for Sanger sequencing at Cosmogenetech.

Editing efficiency was calculated by analyzing each Sanger peak height at the target site[22] using the Indigo software (https://www.gear-genomics.com/indigo/).

## Flow cytometry

To assess the C-to-U RNA editing efficiency using the GFP indicator, $3 \times 10^5$ HEK 293 T cells were seeded in 6-well plates (SPL), incubated for 24 hours, and co-transfected with GFP indicator and padCas13 editor component plasmids using Lipofectamine LTX (Invitrogen). Transfected cells were analyzed using a BD LSRFortessa after 24 hours of light illumination. Single live cells were gated from the whole cell population and used for fluorescence analysis (Supplementary Fig. 15). The flow cytometry results were analyzed with FlowJo X (v. 10.0.7), and the GFP$^+$ proportion of cells was taken as reflecting the editing efficiency.

## Live-cell imaging and electronics

Live-cell imaging was conducted using a Nikon A1R confocal microscope affixed to a Nikon Ti body. The objective was a Nikon CFI Plan Apochromat VC 60X/1.4 numerical aperture (NA), and the Nikon digital zoom imaging software (NIS-elements AR 64-bit version 3.21) was used. A Chamlide TC system (Live Cell Instruments) placed on a microscope stage was used to maintain the live cell conditions at 37°C and 10% $CO_2$.

For light stimulation for 24 hours, a customized 488nm-LED array was used to ensure even blue-light stimulation across all wells. Light intensity was measured at the surface of the LED pad using the photodetector of a PM120D power meter (Thorlabs).

## Imaging processing and analysis

Images were analyzed using Nikon imaging software (NIS-elements AR 64-bit version 5.21) and the ImageJ software. Fluorescence intensity was quantified by integrating densities with the Nikon imaging software. Colocalization between fluorescence channels was quantified, and Pearson's correlation coefficient was applied using the 'Colocalization' tool in the Nikon imaging software. For quantitative analysis of fluorescence intensity in the areas subjected to spatio-temporal regulation, GFP and BFP intensities were measured using the 'Plot profile' tool in ImageJ and normalized on a scale from 0 to 1.

## In vivo light activation and imaging

All animal experiments and procedures complied with the Institutional Animal Care and Use Committees (IACUC) guidelines at the Korea Advanced Institute of Science and Technology (KAIST). The approval number for this study is KA2023-089-v1. Six-week-old male BALB/c were purchased from Raonbio Service (Yongin-si, South Korea). Mice were hydrodynamically transfected by co-injecting padCas13 editor plasmids with a luciferase reporter that encoded the following in sequence: nMagHigh1-dC350-ADAR2$_{DD}$ (E488Q/T490A) mutant, dN351-pMag, firefly luciferase-targeting crRNA, and the luciferase reporter. Transfection was performed using TransIT-EE Hydrodynamic Delivery Solution (Mirus Bio). The total amount of DNA was 150 µg per mouse. Before injection, the abdominal surface fur was removed from mice using a chemical depilatory cream. After injection, the mice were randomly allocated to the dark group or the blue-light group. Mice of the blue-light group were exposed to a - 3 mW/cm$^2$ LED light source (473 nm) located at the bottom of the cage for 16 hours, a timeframe established as effective in previous research[41]. The dark-group mice were placed in a cage that lacked blue-light exposure.

Bioluminescence imaging of mice was conducted 16 hours later. Before luminescence imaging, each mouse was i.p. injected with 200 µL of 15 mg/ml D-luciferin in DPBS. Mice were anesthetized with isoflurane, placed on an IVIS Lumina (Xenogen), and imaged with a CCD camera. The exposure time was set to 1 minute and the data binning levels were set to high. Following in vivo bioluminescence imaging, the liver was isolated from each mouse using standard protocols. The isolated liver was then immersed in a PBS solution containing 2 mM D-luciferin in 12-well culture plates (SPL). Subsequent imaging was performed using an IVIS Lumina (Xenogen) with a CCD camera. To ensure consistency across all samples, the exposure time was uniformly set to 1 minute, and the data binning levels were set to the middle. A region of interest (ROI) was selected for quantification, and the output flow data were analyzed using Image Studio (LI-COR). The ROI was chosen to cover the full signal area where the signal was detected, and a rectangle represented it. The dimensions of this rectangle were kept consistent for all mice and livers in each group to maintain uniformity. Subsequently, the data output from Image Studio was analyzed to obtain the signal intensity. The signal was calculated using the following equation: Signal = Total Intensity for ROI– (Background Mean Intensity x Area (pixels) for ROI).

## Serum biochemistry

Mouse blood was collected using a heart puncture in a serum collection tube and incubated over 1 hour at Room temperature. The serum was separated from blood by centrifugation at 3000 g for 20 minutes at 4 °C and stored at −80 °C until testing. The biochemistry assay was measured by KP&T technology (Osong, South Korea).

## Histology

For paraffin histology, liver tissues were collected and fixed with 4% PFA. The tissue sections were paraffin-embedded, and hematoxylin and eosin (H&E) were stained for pathology.

## Statistics and reproducibility

All experiments were performed at least 3 biologically independent experiments, the results are shown as mean s.em. GraphPad Prism was used for plotting and graphing and Excel was used for statistical analysis.

**Reporting summary**

Further information on research design is available in the Nature Portfolio Reporting Summary linked to this article.

## Data availability

The data supporting the findings of this study are available in the paper and its Supplementary Information. Source Data are provided as a Source Data file. Source data are provided with this paper.

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

## Acknowledgements

This work was supported by the Samsung Science and Technology Foundation under Project Number SSTF-BA1902-06 (to J.Y.[1], J.Y.[2], J.K., and W.D.H.) and the Bio&Medical Technology Development Program of the National Research Foundation (NRF) funded by the Korean government (MSIT) (No. RS-2023-00263628, J.Y.[1], J.Y.[2], J.K., and W.D.H.). Cartoons in Fig. 6a and Supplementary Fig. 11a were created with BioRender.com. [1]Jeonghye Yu [2]Jihwan Yu.

## Author contributions

J.Y.[1] and W.D.H. conceived the project and directed the work. J.Y.[1] designed the experiments. J.Y.[1], J.S., J.Y.[2], and J.K. performed the experiments. J.Y.[1], D.Y., and W.D.H. discussed the data. J.Y.[1] and W.D.H. wrote the manuscript. [1]Jeonghye Yu [2]Jihwan Yu.

## Competing interests

J.Y. and W.D.H. are inventors on a patent application related to this work filed by the Korea Advanced Institute of Science and Technology (PCT/KR2022/009993, filed 08 July 2022). All other authors declare that they have no competing interests.
