## [Peer Review File · Nature Communications]

REVIEWER COMMENTS

Reviewer #1 (Remarks to the Author):

In this interesting study, Jeonghye Yu et al. developed a light-inducible Cas13 system (paCas13) for endogenous transcript degradation, RNA base-editing, and adjustment of post-translational modifications. Starting from PspCas13b structure prediction by Alphafold2 and split site screening, the authors provided sufficient evidence for chemical- or light-induced reassembly as well as functional restoration of split PspCas13b variants. They also demonstrated the proof-of-concept RNA editing with padCas13 editor in mice. However, a similar split Cas13b system (also based on PspCas13b) was published by Ying Xu et al. in JACS (PMID: 36811465). In the JACS article, Ying Xu et al. demonstrated similar results of RNA degradation by abscisic acid (ABA) induced reassembly of padCas13b. Interestingly, by using a synthetic photocaged ABA (ABA-DMNB), Ying Xu et al. enabled light control of the system. Although different split sites of the Cas13b have been tested this manuscript, the level of conceptual and technical innovation remain rather moderate, particularly given the similar efficiency in RNA degradation among these similar studies. To outperform the JACS report, Jeonghye Yu et al. should have exploited the strength of the optogenetic module (e.g., high precision with spatiotemporal control), but these data are not provided in the current study. Overall, Jeonghye Yu et al. provided solid data to support the development of their light-inducible Cas13 system (paCas13) system. However, the novelty is greatly reduced by two earlier reports adopting a similar engineering approach (though using different stimuli to trigger the functional restoration). Unfortunately, the power of paCas13 has not been fully demonstrated in terms of spatial and temporal control in real world applications.

Several issues should be addressed to improve the manuscript, as listed below.

1. Multiple data and figures showing that FKBP-FRB acts as an efficient tool to induce RNA editing in the manuscript. However, they only emphasized their photoactivatable tool. The authors should explain why rapamycin and light can only induce about half RNA degradation, while the intact Cas13b can degrade most of target RNA as shown in Figure 1e,1f, 2b and 2c.
2. In Fig. 2b, please clearly indicate the reference of “normalized luciferase”. In Fig. 2d (also in lines 156-158), the data is insufficient to support higher expression level of C350 fragment than the C352 fragment. In Fig. 2e, positive controls of intact PspCas13b with targeting crRNA should be provided to better compare the effect of the designed split system(s).
3. Similarly in Fig. 3b-e, please indicate the Y-axis clearly in the figure legends (or main text). Also, in Fig. 3d, signals at more time points after light removal (from 12 to 24 hours) should be provided to illustrate temporal regulation of the tool.
4. In Fig. 4, I wonder if the C-to-U editing could also be reported by luciferase assay? I am also curious about the comparison between padCas13 editor and intact PspCas13b.

5. In Fig. 5c, the schematic representation is confusing. Please redraw it. Also, results from intact PspCas13b should be provided.

6. For the in vivo application, the mice were exposed to blue light. It remains vaguely described with regard to how long the mice should be exposed and whether this long-time blue light exposure will cause any detrimental side effects to the mice. Considering blue light cannot penetrate deeply into living tissues (<1 mm), how can blue light work so efficiently as described in this manuscript.

Reviewer #2 (Remarks to the Author):

In the submitted manuscript, the authors reported the development of inducible Cas13-based enzymes that rely either on FKBP-FRB (inducible by rapamycin) or the Magnet system (inducible by light). Switchable DNA/RNA engineering enzymes are useful in many applications and have been pursued by multiple groups in recent years. Conceptually, the splitting of a Cas enzyme is not new (e.g., PMID: 25643054). In fact, there has been a recent publication on a related split Cas13b system for transcriptome engineering as well (PMID: 36811465). Hence, it would be useful if the authors could better clarify how their work is different/unique and potentially useful to other researchers.

Major comments:

1) Is it really necessary to use AlphaFold2, other than it being a hot topic? For example, one can examine the PbuCas13b structure, identify solvent-exposed loops, and then align PspCas13b to map the corresponding residues.

2) Figure 1 - The overall impression I'm getting is that the switch-like behaviour of even the best construct is not great, as degradation is only partial in the presence of inducer. Is there any way to optimize the construct further?

3) Figure 2b - Controls with wildtype (full-length, unsplit) enzyme are missing.

4) Figure 2d - Does the N-terminal fragment change localization upon induction? There should be greater recruitment to cytoplasm in the presence of rapamycin/light.

5) Figure 2e - Please show controls, especially original unsplit enzymes. Additionally, the authors wrote "we observed significantly better light-induced RNA interference using paCas13-1 and -2 ..." Better than what? Also, switching behaviour hardly exists for PPIB. I suggest evaluating the constructs on more target genes.

6) What are the kinetics of degradation (on and off rates, similar to what has been shown for editing)?

7) Does the behaviour of the authors' split RNA editors resemble that of the original REPAIR system? E.g., can ADAR2DD be fused to the N-terminus of the N-fragment? Are guide design rules similar (same mismatch distance etc)?

8) Figure 3f - Please quantify editing rates from the chromatograms.

9) Figure 3 - Positive control (original REPAIR system) is missing.

10) Figure 5a, b - The practical purpose of having an inducible system targeting KRAS/ PPIB is unclear. Without inducer, there is already some editing. Is the 5-10% increase going to make a difference? Why not just use REPAIR?

11) Figure 6 - For the in vivo demonstration, please show some endogenous targets as well.

12) If the analysis of surface interaction of split fragments using MaSIF is useful in ruling out split sites that will lead to auto-reassembly of the enzyme, then the authors should build this into their workflow. Choose other internal positions to divide Cas13b into two, use MaSIF to predict, and then validate comprehensively (both positive and negative cases). If the workflow is accurate, then this manuscript becomes more interesting as other researchers can use it to split their Cas enzymes of interest.

13) Please do some comparison with published work, such as PMID: 36811465.

Minor comments:

14) 351/350 - What does this mean? Is there a duplication of 1-2 amino acids (350-351)? Perhaps the authors can use a schematic to illustrate.

15) Discussion - The authors wrote that "the propensity of auto-assembly increased toward the C-terminal region". This statement is too sweeping, as only a few split sites were examined.

16) Related to comment #12 above, the manner in which the results section and the discussion section were written does not match up. In the results section, the authors used MaSIF after testing their constructs to come up with a potential explanation to explain their data. But in the discussions, the authors wrote that they "found that the interface structure of the auto-assembled split site (N550/C551) was stronger than that of the selected split site (N351/C352); we then validated this experimentally" (which sounds like they made predictions, and then proved their predictions).

17) In some parts of the manuscript, the writing can be clearer and more concise.

Reviewer #3 (Remarks to the Author):

The authors of "Programmable RNA base editing with photoactivatable CRISPR-Cas13" have developed a light-inducible Cas13 system called paCas13 that allows for the manipulation of RNA in vitro and in vivo via split protein designs. The padCas13 editor, a light-inducible base-editing system, was also developed, which enables reversible RNA editing under light and was effective in editing A-to-I and C-to-U RNA bases. The paCas13 system has the potential for broad applications in manipulating RNA in different disease states and physiological processes, offering new avenues for research and therapy.

This reviewer believes that the manuscript should be accepted for publication after the following revisions.

1. The advantage of the split-Cas13 strategy should be discussed as compared to other means of conditional control of CRISPR/Cas functions, for example optically/chemically caged guide RNAs, split inteins, and split effector domains (in this case, split ADAR2DD). Such discussion would provide solid grounds for the authors' design and application of paCas13.

2. The logic presented in the paragraph (line 35-43) is not self-coherent. Smaller-sized Cas13 is not necessarily related to inducible Cas13 systems. To this reviewer's understanding, it was the authors' goal to miniaturize their paCas13 system or to compare with AAV-packaged Cas13s in this manuscript. Please re-write the paragraph to provide reasoning for inducible systems, especially optical control. Deliverability was very well discussed in line 56. Maybe authors can move their discussion for compact Cas13 and AAV here.

3. Why was rapamycin-inducible dimerization explored? Was it chosen to inform optical control because it is a platform that allows for more high-throughput screening? Please provide more logic connection between chemogenetic control and optogenetic control.

4. AF2 is not yet reliable in the prediction of RNP structures. RNA-guided RNA nucleases undergo major structural change upon acquisition of the crRNA (e.g., PMID 30917325, 35643083). The authors should acknowledge this in their explanation for split-design principles. Also, exactly how the 12 iterations of PspCas13 structure was done and what pLDDT was deemed satisfactory should be detailed in methods.

5. This reviewer believes that the targeting of 3'UTR was to accelerate luciferase mRNA degradation. However, the reason for this choice should be pointed out to the broad readership of Nature Communications.

6. Numbering of split sites follows two patterns (single number and N/C double numbers). Please specify their meanings as readers may not be well versed in split protein designs.

7. For spontaneous reconstitution of split Cas13 with split Fluc, was it guide-dependent? Please elaborate and provide information on what plasmids were transfected for each different experiment in experimental methods.

8. No negative controls for +488 nm light but –transfection of plasmids was included. This is a major issue with almost all the experiments in this manuscript. Short-wavelength light can induce the activation of multiple stress pathways as well as cross-talking cell growth pathways, which include KRAS, PPIB, and STAT3, and thus lead to transcriptome interference. A more solid negative control would be +488 nm light with the transfection of split Cas13 (without Magnet domains). Also, cell viability should be verified, too. The authors should seriously consider adding these controls before this manuscript can be published.

9. Please provide potential explanations for nucleus localization of N-term fragment of Cas13. (Fig. 2d)

10. The paragraph between lines 190 and 206 should be rewritten to separate time-course study for activation and time-course study for reversibility. Right now Fig. 3c, e, f are for activation and 3d is for reversibility. For easier comprehension, please put luciferase activation and increase in corrected % by sequencing side-by-side. Then put luciferase decrease (current Fig 3d) and decrease in % editing by sequencing (authors should include this missing piece of data) side-by-side.

Also, Fig. 3c says “normalized luciferase” for the y-axis. Please explain what value the timepoints were normalized to. This reviewer would assume timepoint 0 h, but it's clearly not the case. The same problem is prevalent throughout the manuscript and should be amended, another example is Fig S5.

11. Please provide details on how mice bioluminescence data was quantified, i.e. selection of area of interest for quantification (superimposed onto all n = 5 mice images for each group).

12. Please provide images for all n = 5 mice for each group in Fig. S9a.

13. In the discussion section, the authors should provide more insights into the potential limitations of the paCas13 system and the challenges that need to be addressed in future studies. This will help readers better understand the scope of the study and its implications for future research.

Other minor modifications include:

1. multiple typos and grammar errors throughout the manuscript, e.g., “CRIPSR” in line 25. These mistakes hinder the comprehension by readers and should be amended before publication.

Point-by-point responses to reviewers' comments

We thank all of the reviewers for their constructive comments, which greatly helped us to identify critical issues and improve our manuscript. We have addressed all of the reviewers' points and conducted the requested experiments. Newly added or revised figures are summarized in the table below. The revised parts of the manuscript are indicated in red.

Figures newly added or revised.	
Main Fig. 1d-f, 2b and c	Modified X- and Y-axis labeling to indicate specific targets and split sites
Main Fig. 2b	Compared with full-length Cas13b as a positive control
Main Fig. 2d	Kinetic evaluation of RNA degradation by paCas13
Main Fig. 2e and f	Light-induced subcellular localizations of paCas13 fused with a fluorescent protein in the same cell; changed HeLa cell for quantification and conducted Pearson correlation analysis
Main Fig. 2g	Evaluated effects of paCas13 on targeting endogenous RNAs with additional targets; used full-length Cas13 with targeted crRNA as a positive control and non-targeting crRNA as a negative control
Main Fig. 3b	Provide information on tiling crRNAs targeting firefly transcripts
Main Fig. 3c	Compared with full-length Cas13b fused with ADAR2 _{DD} (wt) using the same tiling crRNAs
Main Fig. 3d	Modified Y-axis labeling to indicate specific targets
Main Fig. 3f	Changed the Y-axis label to "Edited mRNA"
Main Fig. 3g	Cell viability tests under light condition
Main Fig. 3h and i	Reversible optogenetic A-to-I RNA editing of the padCas13, both protein and RNA
Main Fig. 4f	Compared with full-length Cas13b fused with evolved ADAR2 _{DD} for C-to-U RNA base editing
Main Fig. 4g-i	Spatial control of padCas13 editor for light-induced C-to-U RNA base editing
Main Fig. 5a and b	Compared with full-length Cas13b fused with hyper-edited ADAR2 and evolved ADAR2; used no transfection as a

	negative control Changed the Y-axis label to “Edited mRNA”
Main Fig. 5c	Updated scheme to show targeting of CTNNB1 leading to Wnt signaling induction
Main Fig. 6a	Bioluminescence imaging scheme with quantification box
Main Fig. 6d	RNA editing level in mouse liver
Supplementary Fig. 1c	Background effect of light stimulation on RNA knockdown efficiency
Supplementary Fig. 1d	Cell viability effects of active Cas13 constructs under dark and light conditions
Supplementary Fig. 2b	AI-based interface model structure of all split sites by MaSIF-site
Supplementary Fig. 2c	Quantification of interface properties across split sites
Supplementary Fig. 2d	Electrostatic property of PspCas13b
Supplementary Fig. 3	Subcellular localization of paCas13 fragments
Supplementary Fig. 4a	Luciferase activity following transfection with W417X mutant of firefly luciferase with or without full-length Cas13
Supplementary Fig. 4b	Background editing effects under crRNA-only conditions
Supplementary Fig. 4c	Restoration of chemical-induced luciferase activity via A-to-I RNA editing with tiled crRNAs
Supplementary Fig. 4d and e	Compared with tiling patterns among full-length, padCas13-1, and padCas13-2
Supplementary Fig. 8	Bioluminescence images of all five mice
Supplementary Fig. 9	Comparative analysis of liver damage under dark and light exposure in mice
Supplementary Table 1	Added additional crRNA spacer sequence for endogenous RNA knockdown
Supplementary Table 2	Added additional primers for qPCR

Reviewer #1

We thank reviewer #1 for their constructive remarks on our work. Based on these comments, we have made the following changes and further improved the manuscript.

Remarks to the Author:

In this interesting study, Jeonghye Yu et al. developed a light-inducible Cas13 system (paCas13) for endogenous transcript degradation, RNA base-editing, and adjustment of post-translational modifications. Starting from PspCas13b structure prediction by Alphafold2 and split site screening, the authors provided sufficient evidence for chemical- or light-induced reassembly as well as functional restoration of split PspCas13b variants. They also demonstrated the proof-of-concept RNA editing with padCas13 editor in mice. However, a similar split Cas13b system (also based on PspCas13b) was published by Ying Xu et al. in JACS (PMID: 36811465). In the JACS article, Ying Xu et al. demonstrated similar results of RNA degradation by abscisic acid (ABA) induced reassembly of padCas13b. Interestingly, by using a synthetic photocaged ABA (ABA-DMNB), Ying Xu et al. enabled light control of the system. Although different split sites of the Cas13b have been tested this manuscript, the level of conceptual and technical innovation remain rather moderate, particularly given the similar efficiency in RNA degradation among these similar studies. To outperform the JACS report, Jeonghye Yu et al. should have exploited the strength of the optogenetic module (e.g., high precision with spatiotemporal control), but these data are not provided in the current study. Overall, Jeonghye Yu et al. provided solid data to support the development of their light-inducible Cas13 system (paCas13) system. However, the novelty is greatly reduced by two earlier reports adopting a similar engineering approach (though using different stimuli to trigger the functional restoration). Unfortunately, the power of paCas13 has not been fully demonstrated in terms of spatial and temporal control in real world applications.

We appreciate the reviewer's comments on the conceptual similarities between our work and that of Ying Xu et al., published in JACS (PMID: 36811465). We agree that both studies employ a split PspCas13b system and achieve RNA degradation. However, we would like to highlight the distinct advantages of our blue light-inducible paCas13 system.

First, the ABA-based system described in the JACS report requires a chemical inducer and thus involves additional and complicated steps for in vivo application. In contrast, our blue light-inducible system uses an LED-enable cage lid and thus involves a much simpler induction process. Moreover, while the system described in the JACS report utilizes UV light for light control, it also requires the exogenous chemical, ABA-DMNB, which is well-known to trigger DNA damage and cellular toxicity that there are two additional chemicals in the JACS-described system. We now present results obtained from a comparative analysis of liver damage

under light conditions *in vivo*, showing that our blue-light system does not induce any sign of toxicity (new **Supplementary Fig. 9**).

Furthermore, to address the lack of spatiotemporal performance, we extended the padCas13 editor for C-to-U RNA editing that converts a BFP signal to a GFP signal in a light-inducible manner. We conducted an additional experiment where half of a given well was exposed to light, and the other half was kept in the dark. Imaging conducted 24 hours later revealed strong GFP fluorescence only on the light-exposed side. As shown in **Fig. R1**, the region exposed to blue light on the left side of the 6-well black-wall plate exhibited a significant GFP fluorescence signal.

We hope this experiment addresses the reviewer's concern regarding demonstrating the system's capacity to undergo spatial regulation. We included the additional data in the revised manuscript (**Fig. R1**, presented in new **Fig. 4g-i** and **Page 16**).

Figure R1. Spatial control of padCas13 editor for light-induced C-to-U RNA base editing

a, Schematic illustration of strategy for directing local activation of light-induced C-to-U RNA editing activity with a photomask.

b, Representative 20X confocal microscopy images of HEK 293T cells after 24 hours of local light stimulation. Scale bar = 1 mm.

c, Quantitative analysis of fluorescence intensity in the imaged area of panel b, performed using the ImageJ software.

Beyond the difference in the utilized inducer, our system differs from that described in the JACS report in our use of a split-site (N351/C350) that represents a robust site with low basal activity and high inducibility. To provide a more rigorous comparison of this aspect, we conducted additional comparative experiments.

Firstly, to provide a more rigorous comparison with the ABA-based system reported in JACS (PMID: 36811465), we targeted the KRAS transcript using the same ABI-PYL1 configuration described in the previous study (**Fig. R2a**). Our experimental results revealed that the knockdown efficiency was comparable with that obtained using full-length Cas13¹, but the DMSO condition exhibited a significant background activity level, as shown in **Fig. R2b**.

To enable a more precise evaluation, we next targeted the $\lambda 2$ sequence for luciferase readout; this provided an exogenous indicator and allowed us to monitor the system's activity more clearly under induced and non-induced states.

Using this strategy, we comprehensively compared our split-site (N351/C350) against the previously published split-site (N761/C762) under various inducers, including ABA, rapamycin (utilizing FKBP-FRB), and blue light (utilizing Magnet), as shown in **Fig. R2c-e**.

We extended the comparative analysis to a direct determination of RNA base editing, which is a crucial aspect of our study. Using the same split sites, we found that the previously published split-site (N761/C762) exhibited a higher background dark-state activity level in RNA base editing (**Fig. R3**).

Our comprehensive comparison thus revealed several key distinctions between the previously reported system and our new system. Notably, the split-site (N761/C762) used in the previously published work exhibited considerable background effects in RNA knockdown (**Fig. R2**) and RNA base editing (**Fig. R3**), neither of which was detailed in the prior study. In contrast, our system using the N351/C350 split site demonstrates high-level inducibility with minimal background noise, allowing for more accurate and reliable data interpretation.

We hope our responses resolve the reviewer's concern.

Figure R2. Comparative analysis of our inducible systems with the previously published split Cas13-761 (PMID: 36811465)

- a, Construction of full-length Cas13b and ABA-inducible system for comparative analysis.
- b, Endogenous KRAS expression level under 100 μ M ABA induction. (n = 3 independent experiments)
- c, ABA-inducible RNA knockdown efficiency. (n = 3 independent experiments)
- d, Rapamycin-inducible RNA knockdown efficiency. (n = 3 independent experiments)
- e, Light-inducible RNA knockdown efficiency. (n = 3 independent experiments)

Figure R3. Comparative analysis of A-to-I RNA base editing under various induction conditions.

- a, Construction of full-length Cas13b and ABA inducible system for comparative RNA editing analysis.

- b, ABA-inducible luciferase restoration mediated by A-to-I RNA base editing. (n = 3 independent experiments)
- c, Light-inducible luciferase restoration mediated by A-to-I RNA base editing. (n = 3 independent experiments)
- d. Chromatogram of RNA base editing at the targeted site.
- e, Quantitative assay of RNA editing levels shown in panel d. (n = 3 independent experiments)

Several issues should be addressed to improve the manuscript, as listed below.

1. Multiple data and figures showing that FKBP-FRB acts as an efficient tool to induce RNA editing in the manuscript. However, they only emphasized their photoactivatable tool. The authors should explain why rapamycin and light can only induce about half RNA degradation, while the intact Cas13b can degrade most of target RNA as shown in Figure 1e,1f, 2b and 2c.

We thank the reviewer for their insightful comment regarding the reduced efficiency of our inducible Cas13b systems compared to the intact Cas13b system (referred to as ‘full-length Cas13’ in the manuscript and figures). This may indicate that the need for additional activation inducers, such as rapamycin or light exposure, constrains the overall activity of the splitting-based system. Furthermore, RNA targeting presents inherent complexities, especially compared to DNA-targeting systems, such as CRISPR-Cas9. While edited DNA remains permanently altered, the targeting of RNA is inherently transient due to the continuous nature of the transcription, splicing, translation, and degradation processes.

Despite the admittedly lower efficiency, however, our inducible systems offer unique advantages by offering spatiotemporal control (**Fig. R1, new Fig. 3, 4 and Supplementary Fig. 4c**), which is necessary for studying dynamic biological processes and allows users to manipulate RNA editing with a level of control that is not offered by intact Cas13b.

We agree that these points should be added to enable the reader to fully understanding our data presented, and we discussed them in the revised manuscript (**Page 21**).

2. In Fig. 2b, please clearly indicate the reference of “normalized luciferase”. In Fig. 2d (also in lines 156-158), the data is insufficient to support higher expression level of C350 fragment than the C352 fragment. In Fig. 2e, positive controls of intact PspCas13b with targeting crRNA should be provided to better compare the effect of the designed split system(s).

We appreciate the reviewer's attention to detail regarding this query on the normalization of our data in Fig. 2b. We would like to clarify that the "normalized luciferase" was calculated as the ratio of firefly luminescence (fluc) to Renilla luminescence (Rluc). We apologize for any confusion caused by the use of "normalized luciferase". In the revised manuscript, we changed this language to avoid any potential misunderstanding (new **Fig. 2b and c**, new **Supplementary Fig. 1**).

Regarding Fig. 2d, we acknowledge that the fluorescence-based measurement may not accurately represent protein expression levels and could be open to subjective interpretation. Therefore, we decided to remove this sentence in the revised manuscript.

Regarding Fig. 2e, the revised version of this figure now includes the comparative data (new **Fig. 2g** (original **Fig. 2e**)).

3. Similarly in Fig. 3b-e, please indicate the Y-axis clearly in the figure legends (or main text). Also, in Fig. 3d, signals at more time points after light removal (from 12 to 24 hours) should be provided to illustrate temporal regulation of the tool.

We apologize for any confusion caused by this oversight in our initial manuscript and appreciate the reviewer's assistance in enhancing the clarity of our study.

Regarding the Y-axis in Fig. 3b-e, as mentioned above, we revised these panels (new **Fig. 3c-i**) to ensure that the Y-axis is accurately and clearly labeled.

As for original **Fig. 3d**, we appreciate the reviewer's suggestion that the inclusion of additional time points would more comprehensively illustrate the temporal regulation of our tool. To this end, we conducted analyses every 3 hours within the specified interval (**Fig. R4a**, presented in new **Fig. 3h**). Our updated results revealed that while padCas13-1 and padCas13-2 exhibit distinct kinetics, they both demonstrate a gradual and reversible transition to the dark state.

Furthermore, we directly analyzed RNA editing levels to offer a more in-depth understanding of this reversibility. Unlike the luciferase signal, which peaked at the 12-hour mark before gradually decreasing, the RNA levels declined steadily to align with the dark state by the 24-hour mark (**Fig. R4b**, presented in new **Fig. 3i**). These findings show that RNA and protein synthesis and translation exhibit different kinetics under the paCas13 condition.

The revised manuscript incorporates these additional data (**Fig. R4**, presented in new **Fig. 3h and i** and on **Page 14**).

Figure R4. Reversible optogenetic A-to-I RNA editing with the padCas13 editor

a, Reversibility of light-induced luciferase restoration by the padCas13 editor was assessed after blue-light illumination was removed. The data obtained at each time point were normalized to those obtained from the corresponding dark-exposed control group. (n = 3 independent experiments)

b, Direct quantification of RNA editing percentages was performed using RT-PCR-Sanger sequencing after blue-light illumination was removed. The data obtained at each time point were normalized to those obtained from the corresponding dark-exposed control group. (n = 3 independent experiments)

4. In Fig. 4, I wonder if the C-to-U editing could also be reported by luciferase assay? I am also curious about the comparison between padCas13 editor and intact PspCas13b.

As the reviewer mentioned, we could use the luciferase assay to measure C-to-U editing, such as by introducing a stop codon (e.g., Q(CAA) to X(UAA)). However, this approach has the significant limitation that residual luciferase proteins generated before the editing might still be detected and, therefore, correction to a stop codon could confound our results by causing an overestimation of editing efficiency. Thus, we chose to employ a fluorescence-based readout system. This approach bypasses the problem of residual protein detection and provides a more reliable means to report on C-to-U editing in our experimental context.

Regarding the suggestion to compare our padCas13 system to intact Cas13b, we added these data to the revised manuscript (new **Fig. 4f**).

5. In Fig. 5c, the schematic representation is confusing. Please redraw it. Also, results from intact PspCas13b should be provided.

We acknowledge the need for a clearer schematic representation and revised this panel accordingly (new **Fig. 5c**). Moreover, we understand the importance of providing results from intact PspCas13b for comparison, and included these data in the revised manuscript (new **Fig. 5**).

6. For the in vivo application, the mice were exposed to blue light. It remains vaguely described with regard to how long the mice should be exposed and whether this long-time blue light exposure will cause any detrimental side effects to the mice. Considering blue light cannot penetrate deeply into living tissues (<1 mm), how can blue light work so efficiently as described in this manuscript.

We appreciate the reviewer's insightful comment on in vivo application. We chose 16 hours for the duration of light exposure because this timeframe was previously reported to be effective², which utilizes the same photoactivatable protein (Magnet system). We revised the Methods section to clarify this point (**Page 30**).

Regarding potential side effects, we acknowledge the reviewer's concerns regarding the long-term impact of blue-light exposure in mice. Our approach relies on LED plate-based noninvasive illumination, which differs from conventional fiber optic systems. In conventional systems, light emerges from a small point and the generated heat could potentially cause side effects. However, our LED plate-based system distributes the light over a broader area with weaker intensity; this reduces the potential for side effects, including thermal effects. However, to evaluate potential side effects, we specifically assessed liver damage, which is often used as a parameter for monitoring systemic toxicity³. In our experimental design, mice were separated into dark and light groups and exposed to experimental conditions that differed only in the light exposure. Blood samples were collected from the hearts of mice from both groups, and the levels of alanine aminotransferase (ALT), aspartate aminotransferase (AST), alkaline phosphatase (ALP), and total bilirubin were measured. Concurrent histological examinations of mouse liver tissues were performed through hematoxylin and eosin (H&E) staining. Our analyses did not reveal any significant alteration of the assessed parameters in either group. These additional results have been integrated into our revised manuscript to provide a more comprehensive understanding of the safety profile and experimental validity of our in vivo application (**Fig. R5**, presented in new **Supplementary Fig. 9** and **Page 19**).

Figure R5. Comparative analysis of liver damage in dark- and light-exposed mice

a, Biochemical analysis of blood samples from livers of mice exposed to dark and light conditions, assessing levels of alanine aminotransferase (ALT), aspartate aminotransferase (AST), alkaline phosphatase (ALP), and total bilirubin. (n = 4 mice per group)

Error bars represent the mean \pm s.e.m.

b, Histological analysis of hematoxylin and eosin (H&E)-stained livers. Images are presented at magnifications of 1x (scale bar = 2 mm), 4x (scale bar = 300 μ m), and 20x (scale bar = 60 μ m).

Regarding the reviewer's concerns on the efficiency limitations of blue-light penetration, we agree that this must be considered in our in vivo experiments. The blue-light-activated PA-Cre recombinase system² has proven effective in multiple studies^{2, 4-7}. In one such study, the enhanced PA-Cre (ePA-Cre) system was used in a stable transgenic mouse model to provide comprehensive data on blue-light penetration across various tissues⁷. Their findings showed that blue light could penetrate up to 750-875 μ m in tissues such as the heart, liver, and kidney when exposed to 48 hours of LED plate-based illumination.

Light-dependent recombination activity⁷ (data from ref 7)

We acknowledge that such constraints on blue-light penetration could inherently limit the in vivo applications of our padCas13-based RNA editing system. We included this concept in the revised manuscript (**Page 18**).

Reviewer #2

We thank reviewer #2 for their critical comments and constructive remarks, which enabled us to significantly improve our revised manuscript. According to the reviewer's comments, we have made the changes and further improved the manuscript, as detailed below.

Remarks to the Author:

In the submitted manuscript, the authors reported the development of inducible Cas13-based enzymes that rely either on FKBP-FRB (inducible by rapamycin) or the Magnet system (inducible by light). Switchable DNA/RNA engineering enzymes are useful in many applications and have been pursued by multiple groups in recent years. Conceptually, the splitting of a Cas enzyme is not new (e.g., PMID: 25643054). In fact, there has been a recent publication on a related split Cas13b system for transcriptome engineering as well (PMID: 36811465). Hence, it would be useful if the authors could better clarify how their work is different/unique and potentially useful to other researchers.

Major comments:

1) Is it really necessary to use AlphaFold2, other than it being a hot topic? For example, one can examine the PbuCas13b structure, identify solvent-exposed loops, and then align PspCas13b to map the corresponding residues.

Indeed, we initially considered solvent-exposed loops in the PbuCas13b structure, as suggested by the reviewer. However, we found that this approach limited us to existing structural data and did not provide sufficiently detailed insights into split Cas13b. AlphaFold2 provided accurate and previously inaccessible structural predictions for PspCas13b and split Cas13b, which we required for our subsequent MaSIF-site analysis.

Moreover, AlphaFold2 is valuable for researchers who are not structural biologists and who rely on experimental screening to interpret molecular interactions. We analyzed all predicted split Cas13 and PspCas13b structures through MaSIF-site and APBS, and used the results to evaluate interface structures and electrostatic properties (**Fig. R6**, presented in new **Supplementary Fig. 2**).

Figure R6. Computational prediction of characterization across the split Cas13 candidates

a, Schematic representation of the prediction task workflow, from AlphaFold2 to MaSIF-site analysis.

b, Visual comparison of split Cas13 candidates. Yellow triangles represent an interface, the yellow box represents split candidates without auto-assembly, and the blue box represents split candidates with auto-assembly.

c, Quantification of interface properties across split sites. The utilized total interface ratio thresholds were 0.5 and above and 0.6 and above. The total interface ratio was calculated based on the number of atoms in each fragment exceeding the given score threshold, normalized by the total number of atoms in both fragments.

d, Electrostatic property of PspCas13b. Charge values above + 30 and below -30 were capped at those values and then normalized between -1 and 1.

2) Figure 1 - The overall impression I'm getting is that the switch-like behaviour of even the best construct is not great, as degradation is only partial in the presence of inducer. Is there any way to optimize the construct further?

We appreciate the reviewer's comment on the need to further optimize our split Cas13 system. We agree that split Cas13 demonstrates only partial degradation. Importantly, however, we found that further enhancing the RNA degradation ability of the split Cas13 system led to increased background activity, which is a well-documented challenge in inducible systems influenced by factors such as leaky promoters or off-target effects⁸. We also experimented with adding nuclear localization signals to improve the efficiency, but found that such modifications led to unwanted background signals, as shown in **Fig. R7**.

We agree that the degradation capacity of the reassembled Cas13 is low in comparison to the intact Cas13. We think this reflects that the need for an additional activation inducer, such as rapamycin or light exposure, constrains the overall activity of the splitting-based system. Furthermore, RNA targeting presents inherent complexities, especially compared to DNA-targeting systems such as CRISPR-Cas9. While edited DNA remains permanently altered, the targeting of RNA is inherently transient due to the continuous nature of the transcription, splicing, translation, and degradation processes.

Despite the acknowledged efficiency differences, however, our inducible systems offer unique advantages in enabling spatiotemporal control (new **Fig. 3, 4 and Supplementary Fig. 4c**), which is essential for the study of dynamic biological processes and allows us to manipulate RNA editing in a controlled manner that cannot be achieved with intact Cas13b.

In addition, as discussed in the manuscript, we are currently exploring the application of directed evolution techniques to improve the efficiency and reduce the background noise of

paCas13. Directed evolution will allow for high-throughput screening and facilitate efficient exploration; it offers the opportunity to generate variants with optimized characteristics that are broadly applicable and not limited by the initial screening conditions. The success of this method has already been demonstrated in specific enzymes, such as compact prime editors⁹ and evolved ADAR2¹⁰, the latter of which was also utilized in this study.

A recent study on optogenetic tools for enzyme optimization¹¹ also suggests some intriguing directions. We are considering applying our tool under light and dark conditions to fine-tune protein function for RNA knockdown or editing with minimal background activity.

We added these limitations to the revised manuscript and expanded the Discussion section to provide a more comprehensive analysis of potential optimization strategies (**Pages 21-22**).

Figure R7. Comparative analysis of split Cas13 efficiency and background noise with adding nuclear localization signals

All values were normalized to those obtained from the corresponding non-targeted (NT) crRNA group under 0nM and 500nM conditions. (n = 3 independent experiments)

3) Figure 2b - Controls with wildtype (full-length, unsplit) enzyme are missing.

In response to this comment, we included control experiments performed using full-length Cas13b in the revised manuscript (new **Fig. 2b**).

4) Figure 2d - Does the N-terminal fragment change localization upon induction? There should be greater recruitment to cytoplasm in the presence of rapamycin/light.

As the reviewer suggested, we further investigated the change in localization of the N-terminal fragment upon induction. Following light stimulation, we observed a significant increase in the correlation between the N- and C-terminal fragments, suggesting greater recruitment of the N-terminal fragment to the cytoplasm. These results indicated that our inducible system functions as designed and further clarified its underlying mechanism. We included these results in the revised manuscript (**Fig. R8**, presented in new **Fig. 2e** and **f** and on **Pages 11-12**).

Figure R8. Subcellular localization of N- and C-terminal fragment upon light stimulation

a, Construction of paCas13 C-terminally fused with fluorescence protein and representative confocal microscopy images of HeLa cells co-transfected with this paCas13. Scale bar = 20 μm.

b, Pearson's correlation analysis of the subcellular co-localization shown in panel a.

5) Figure 2e - Please show controls, especially original unsplit enzymes. Additionally, the authors wrote "we observed significantly better light-induced RNA interference using paCas13-1 and -2 ..." Better than what? Also, switching behaviour hardly exists for PPIB. I suggest evaluating the constructs on more target genes.

As the reviewer requested, we carried out additional experiments to demonstrate the efficiency of our system and provide a more comprehensive analysis.

We conducted endogenous knockdown experiments and compared the results to those obtained with full-length Cas13b, as a positive control. We also performed additional tests on more target genes and compared the efficiency of our system against that of full-length Cas13 for each gene. These new results have been included in the revised manuscript (**Fig. R9**, presented in new **Fig. 2g**).

Regarding the switching behavior, we acknowledge that the initial experimental setup targeted endogenous RNAs with multiplexed crRNAs. In our initial design, the threefold higher

transfection of multiplexed crRNAs led to a relative reduction in the molar concentration of the paCas13 plasmid. This imbalance resulted in suboptimal expression of paCas13, affecting its RNA degradation efficiency. To address this, we transitioned to using a single-plasmid setup that offers a more balanced molar ratio between the crRNAs and paCas13, and thus allows for a more accurate assessment of light-induced effects on each target (**Fig. R9**, presented in new **Fig. 2g**).

Regarding the statement that “we observed significantly better light-induced RNA interference using paCas13-1 and -2 ...” we sincerely apologize for the confusion caused by this wording. We revised this statement to avoid confusion (**Page 12, Lines 213-214**).

Figure R9. Comprehensive evaluation of endogenous transcript targeting ability of paCas13

Light-induced RNA knockdown of endogenous transcripts in HEK 293T cells transfected with the full-length Cas13b or paCas13 candidates with non-targeted (NT) or targeted crRNAs. All values were normalized to the corresponding NT crRNA group within each dark and light condition. (n = 3 independent experiments)

6) What are the kinetics of degradation (on and off rates, similar to what has been shown for editing)?

As the reviewer suggested, we conducted experiments to examine the kinetics of degradation (**Fig. R10**). The results showed that the paCas13 system exhibited the most pronounced RNA degradation effects at 24 hours post-light illumination and that, notably, the degradation kinetics varied between the two constructs, with paCas13-2 displaying a faster and more stable RNA degradation profile than paCas13-1. We included this information in the revised manuscript (**Fig. R10**, presented in new **Fig. 2d** and on **Page 10**).

Figure R10. Kinetic evaluation of RNA degradation by paCas13

Time-course analysis of paCas13 activity in the presence of light, as determined by the luminescence fold change normalized to that of each dark group. (n = 3 independent experiments)

7) Does the behaviour of the authors' split RNA editors resemble that of the original REPAIR system? E.g., can ADAR2DD be fused to the N-terminus of the N-fragment? Are guide design rules similar (same mismatch distance etc)?

Conceptually, our padCas13 editor is a split and reassembled version of the original REPAIR system. We considered multiple aspects in seeking to clarify the RNA editing behavior of our system compared to the original REPAIR system.

Firstly, our constructs are designed using the same ADAR2_{DD} configuration (C-terminal to the Cas13b protein) and same crRNA (same mismatch distance) as in the original REPAIR system. We found that although there were between-system differences in the absolute RNA editing efficiency (new **Fig. 3c**), the editing behavior patterns were highly correlated (Pearson r correlation values were 0.76 and 0.91) when normalized on a 0-1 scale (**Fig. R11c**).

Next, in the context of PspCas13b, it is known that the N-terminal HEPN1 domain is crucial for RNA targeting and editing¹, and the major of 40% homology in the N-terminus with PbuCas13b¹². Thus, we focused on different frames for fusion rather than fusing to the conserved functional regions. Accordingly, we investigated the RNA editing patterns obtained using engineered crRNAs designed to bind MS2 coat protein (MCP). The engineered crRNAs were transfected with MCP-ADAR2_{DD} fused at a non-C-terminal location (**Fig. R11a**).

Interestingly, the RNA editing patterns observed in this setup differed significantly from those obtained when ADAR2_{DD} was fused at the C-terminus. The most efficient editing was obtained at a mismatch distance of 46, contrasting with the distance of 40 observed for the original C-terminal fusions (**Fig. R11b**). Further, comparing these results to those obtained with the C-terminal fusions of full-length Cas13 and split Cas13 yielded much lower Pearson r correlation values (0.41 and 0.48) than that of padCas13 editor (**Fig. R11c**). These findings demonstrate

that our padCas13 editors exhibit some unique behaviors while maintaining certain similarities with the original REPAIR system.

To avoid confusion about using crRNAs and clarify the RNA editing behavior of the padCas13 editor, we included these results and schematics explaining the tiling crRNAs in our revised manuscript (new **Fig. 3b**, new **Supplementary Fig. 4**, and on **Pages 13-14**).

Figure R11. Comparative analysis of RNA editing behavior obtained using standard and engineered crRNAs

a, Schematic diagram of engineered crRNA designed for MS2 coat protein (MCP) binding and MCP-ADAR2.

b, Normalized relative luminescence (Scaled 0-1) representing RNA editing patterns across various tiling assays.

c, Pearson r correlation analysis of the RNA editing patterns presented in panel b.

8) Figure 3f - Please quantify editing rates from the chromatograms.

We apologize for the confusion caused by the sequencing of the figures. Quantification of the editing rates from the chromatograms in original **Fig. 3f** (new **Fig. 3e**) is provided in original **Fig. 3e** (new **Fig. 3f**). To prevent further confusion, we adjusted the order of these panels in the revised manuscript (new **Fig. 3e and f**).

9) Figure 3 - Positive control (original REPAIR system) is missing.

To address this comment, we conducted the necessary experiments and now present a comparison with positive control data in the revised manuscript (new **Fig. 3c**).

10) Figure 5a, b - The practical purpose of having an inducible system targeting KRAS/ PPIB is unclear. Without inducer, there is already some editing. Is the 5-10% increase going to make a difference? Why not just use REPAIR?

We appreciate the reviewer's attention to detail regarding the RNA editing level in the absence of an inducer. These targets were selected to align with those used in previous studies^{1, 13, 14} and thereby enable the padCas13 editor to be more directly compared with established benchmarks. We now include full-length Cas13 controls for A-to-I and C-to-U RNA editing using non-targeted (NT) and target-specific crRNAs (**Fig. R12**, presented in new **Fig. 5**) to enable direct comparison with the REPAIR system.

The padCas13 editor showed a statistically significant increase in RNA editing activity under the light condition compared to the dark condition. Full-length Cas13 and a NT crRNA showed slight increases in editing efficiency in the dark. We also included a "no transfection" (no TF) control to account for basal editing levels that are naturally present in cells (**Fig. R12**, presented in new **Fig. 5**).

Regarding the 5-10% increase in RNA editing, although it may appear modest, it is statistically significant and comparable to that obtained with full-length Cas13. This indicates that the padCas13 editor, while slightly less efficient than full-length Cas13, still exhibits meaningful RNA editing activity under light exposure. We now address these critical points in the revised manuscript (new **Fig. 5 and Page 17**).

Figure R12 (presented in new **Fig. 5**). Inducible editing of endogenous transcripts by the padCas13 editor compared with full-length Cas13

a, RT-PCR-Sanger sequencing-based quantification of A-to-I RNA editing by full-length Cas13b or padCas13-2 with E488Q/T490A mutation of ADAR2_{DD} after 24 hours of light stimulation. Groups are no transfection (no TF) and, for full-length Cas13b, non-targeted (NT) and targeted (T) crRNAs. (n = 4 or 3 independent experiments)

b, RT-PCR-Sanger sequencing-based quantification of C-to-U RNA editing by the full-length Cas13b or padCas13-2 with evolved ADAR2_{DD} after 24 hours of light stimulation. Groups are as described in panel a. (n = 3 independent experiments)

c, Quantification of β -catenin reporter activation by padCas13 editors in the presence of light. For full-length Cas13b, groups are non-targeted (NT) and targeted (T) crRNAs. (n = 3 independent experiments)

11) Figure 6 - For the in vivo demonstration, please show some endogenous targets as well.

We appreciate the reviewer's suggestion that we include in vivo demonstrations with some endogenous targets to more comprehensively validate the padCas13 editor system. However, we encountered considerable technical challenges in implementing this work.

Unlike our findings when we used adeno-associated viral (AAV) delivery, tail-vein injection of the padCas13 editor system to mice yielded uneven expression across hepatocytes, limiting our ability to assess endogenous RNA editing efficiency (**Fig. R13**). To address this, we tagged the C-termini of the full-length and N-fragment component of padCas13 with mCherry via P2A to enable expressing cells to be sorted. Despite using a standard two-step collagenase perfusion and 40% Percoll with low-speed centrifugation¹⁵, we were unable to successfully sort hepatocytes based on mCherry fluorescence. This failure could be attributed to experimental conditions and/or the intrinsic challenges of tail-vein injection, which involves a hydrodynamic system with rapid delivery of nucleic acids to the liver within 5 seconds¹⁶.

In addition, the conventional protocol for TransIT-EE hydrodynamic delivery solution (Mirus) indicates that gene expression peaks between 8-24 hours post-injection but declines rapidly thereafter. This also limited our ability to measure sustained endogenous editing.

We acknowledge that the padCas13 editor described in our study is in its initial stages of development. Not unlike the PA-Cre system, which required multiple published rounds of optimization^{2, 4-7}, our padCas13 system will require further refinement for robust in vivo applications.

Figure R13. Heterogenous hepatocyte expression of luciferase reporter

a, Luminescence images of mice bearing the luciferase reporter.

b, Luminescence images of whole livers collected from the mice shown in panel a.

c, Luminescence images of liver sections derived from the whole livers shown in panel b.

As an alternative strategy for addressing the reviewer’s comment, we focused on demonstrating the efficiency of the padCas13 editor in a mouse model using an exogenous firefly luciferase W417X reporter. Mice were grouped under “Dark” or “Light” conditions, and RNA editing efficiency was assessed following light stimulation. We believe our findings provide valuable proof-of-concept for the light-dependent function of the padCas13 editor in an in vivo model. We included data on the RNA editing level in the mouse model and discussed the further optimization of the padCas13 editor in the revised manuscript (**Fig. R14**, presented in new **Fig. 6d**).

Figure R14. Quantitative evaluation of RNA editing by the padCas13 editor in an in vivo mouse model

RT-PCR-Sanger sequencing-based quantification of A-to-I RNA editing by the padCas13-2 editor after 16 hours of light stimulation. (n =3 mice per group)

12) If the analysis of surface interaction of split fragments using MaSIF is useful in ruling out split sites that will lead to auto-reassembly of the enzyme, then the authors should build this into their workflow. Choose other internal positions to divide Cas13b into two, use MaSIF to predict, and then validate comprehensively (both positive and negative cases). If the workflow is accurate, then this manuscript becomes more interesting as other researchers can use it to split their Cas enzymes of interest.

We appreciate the reviewer's insightful guidance and thank them for pointing out the potential advantages of integrating a MaSIF-site analysis into our workflow. In our initial manuscript, we explored a variety of split candidates but did not employ the MaSIF-site tool to predict the potential auto-assembly of all the split fragments. As the reviewer suggested, we analyzed additional split-fragment surface interactions with MaSIF-site tool (**Fig. R15**, presented in new **Supplementary Fig. 2**).

Upon conducting the MaSIF-site analysis and digitizing the protein structures into numerical data, we observed a potent alignment between the experimental and computational results, which reinforces the utility of MaSIF-site in our structural interpretation (**Fig. R15b** and **c**). This use of an integrated approach increases confidence in the constancy of our method and enhances the applicability of our study for other researchers interested in splitting their proteins. We described this result in the revised manuscript (**Pages 7-8**).

That said, we would like to highlight certain limitations of this computational comparison. Our study focused exclusively on the split fragments of the PspCas13b protein. While our findings hold within this specific context, it may not be straightforward to extrapolate these results to other proteins. Our study provides valuable insights and a potentially useful framework, but additional research would be required to adapt this workflow for other proteins or Cas enzymes. We included this limitation of the prediction strategy in the revised manuscript (**Page 7** and **Lines 122-125**).

Figure R15 (Same as **Fig. R6**). Computational prediction of characterization across the split Cas13 candidates

a, Schematic representation of the prediction task workflow, from AlphaFold2 to MaSIF-site analysis.

b, Visual comparison of split Cas13 candidates. Yellow triangles represent an interface; the yellow box represents split candidates without auto-assembly; and the blue box represents split candidates with auto-assembly.

c, Quantification of interface properties across split sites. The utilized total interface ratio thresholds were 0.5 and above and 0.6 and above. The total interface ratio was calculated based on the number of atoms in each fragment exceeding the given score threshold, normalized by the total number of atoms in both fragments.

d, Electrostatic property of PspCas13b. Charge values above + 30 and below -30 were capped at those values and then normalized between -1 and 1.

13) Please do some comparison with published work, such as PMID: 36811465.

As the reviewer suggested, we compared our system with that described in the cited report, and observed distinct differences (PMID: 36811465). For example, the split-site (N761/C762) utilized in the previous study tended to auto-assembly, although this was not explored in the publication. The chosen split-site (N761/C762) yielded good inducibility, but experiments performed using this split site in our setup revealed substantial background signals in RNA knockdown and RNA base editing (**Fig. R16 and R17**). This background noise could potentially mask the actual effect of RNA modulation, and could lead to overestimation of the impact for inducibility.

We conducted additional direct comparisons and obtained results that highlighted the robustness and low background noise of our chosen split-site (N351/C350). Firstly, we targeted the KRAS transcript using the ABI-PYL1 configuration outlined in the previous study (**Fig. R16a**). The knockdown efficiency was comparable to that obtained with full-length Cas13¹, but there was a significant background activity level in the DMSO condition, as shown in **Fig. R16b**.

We then targeted the $\lambda 2$ sequences using a luciferase readout, which provided an exogenous indicator that allowed us to more clearly monitor the system's activity in induced and non-induced states.

Using this strategy, we comprehensively compared our split-site (N351/C350) against the previously published split-site (N761/C762) in the presence of various inducers, including ABA, rapamycin (utilizing FKBP-FRB), and blue light (utilizing Magnet).

We further extended the comparative analysis to RNA base editing, which is a critical aspect of our study. We found that the previously published split-site (N761/C762) exhibited high background activity levels in RNA base editing. Moreover, direct quantification of RNA levels revealed elevated editing levels in the dark state compared to that obtained with our N351/C350 split site (**Fig. R17**).

In summary, our comprehensive comparison revealed several key distinctions between our system and the previously published system. Notably, the split-site (N761/C762) utilized in the prior work exhibited considerable background effects in RNA knockdown (**Fig. R16**) and RNA base editing (**Fig. R17**), neither of which was explored in detail in the published study. In contrast, our system (N351/C350) demonstrates high-level inducibility with minimal background noise, allowing for more accurate and reliable data interpretation.

Figure R16 (Same as **Fig. R2**). Comparative analysis of our inducible systems with the previously published split Cas13-761 (PMID: 36811465)

- a, Construction of full-length Cas13b and ABA-inducible system for comparative analysis.
- b, Endogenous KRAS expression level under 100 μ M ABA induction. (n = 3 independent experiments)
- c, ABA-inducible RNA knockdown efficiency. (n = 3 independent experiments)

d, Rapamycin-inducible RNA knockdown efficiency. (n = 3 independent experiments)

e, Light-inducible RNA knockdown efficiency. (n = 3 independent experiments)

Figure R17 (Same as **Fig. R3**). Comparative comparison of A-to-I RNA base editing under various induction conditions.

a, Construction of full-length Cas13b and ABA-inducible system for comparative RNA editing analysis.

b, ABA-inducible luciferase restoration mediated by A-to-I RNA base editing. (n = 3 independent experiments)

c, Light-inducible luciferase restoration mediated by A-to-I RNA base editing. (n = 3 independent experiments)

d. Chromatogram of RNA base editing at the targeted site.

e, Quantitative assay of RNA editing levels shown in panel d. (n = 3 independent experiments)

Minor comments:

14) 351/350 - What does this mean? Is there a duplication of 1-2 amino acids (350-351)? Perhaps the authors can use a schematic to illustrate.

The notation “351/350” refers to the protein being split at the amino acids N351 and C350 (new **Fig. 2c**). We understand this could be confusing and agree that a schematic would help illustrate this more clearly. To address this, the numbering scheme has been standardized across all figures and the manuscript text using the N/C double numbers. This format allows for the specification of adjacent amino acids (e.g., N351/C352) and non-adjacent or overlapping residues (e.g., N351/C350). A clear explanation of this numbering scheme has been included in the Methods section (**Page 23**) to ensure clarity for all readers.

15) Discussion - The authors wrote that "the propensity of auto-assembly increased toward the C-terminal region". This statement is too sweeping, as only a few split sites were examined.

We understand that our observation was based on a limited set of split sites and may not be broadly applicable. To address this point and avoid any potential misunderstanding, we removed the sentence.

16) Related to comment #12 above, the manner in which the results section and the discussion section were written does not match up. In the results section, the authors used MaSIF after testing their constructs to come up with a potential explanation to explain their data. But in the discussions, the authors wrote that they "found that the interface structure of the auto-assembled split site (N550/C551) was stronger than that of the selected split site (N351/C352); we then validated this experimentally" (which sounds like they made predictions, and then proved their predictions).

We apologize for any confusion our manuscript may have caused regarding the sequence in which our experiments were conducted. The computational analysis was performed after our initial cell-based experiments. We aimed to utilize the results of the computational analysis to explore the differences in the auto-assembly of split sites N550/C551 and N351/C352. Considering this, we rewrote the paragraph to clarify our meaning in the revised manuscript (**Page 20 and Lines 371-375**).

17) In some parts of the manuscript, the writing can be clearer and more concise.

We enlisted native speakers to check the manuscript.

Reviewer #3

We are grateful to reviewer #3 for their positive assessment of our work and constructive remarks. To address their comments, we have made changes and further improved the manuscript, as detailed below.

Remarks to the Author:

The authors of “Programmable RNA base editing with photoactivatable CRISPR-Cas13” have developed a light-inducible Cas13 system called paCas13 that allows for the manipulation of RNA in vitro and in vivo via split protein designs. The padCas13 editor, a light-inducible base-editing system, was also developed, which enables reversible RNA editing under light and was effective in editing A-to-I and C-to-U RNA bases. The paCas13 system has the potential for broad applications in manipulating RNA in different disease states and physiological processes, offering new avenues for research and therapy.

1. The advantage of the split-Cas13 strategy should be discussed as compared to other means of conditional control of CRISPR/Cas functions, for example optically/chemically caged guide RNAs, split inteins, and split effector domains (in this case, split ADAR2DD). Such discussion would provide solid grounds for the authors’ design and application of paCas13.

We appreciate the reviewer’s insightful suggestion that we compare our split Cas13 strategy with other conditional control methods. We included this discussion in our revised manuscript (**Page 20**) and agree that this material provides a solid foundation for our design and validates the broad potential utility of paCas13 in RNA research and therapy.

2. The logic presented in the paragraph (line 35-43) is not self-coherent. Smaller-sized Cas13 is not necessarily related to inducible Cas13 systems. To this reviewer’s understanding, it was the authors’ goal to miniaturize their paCas13 system or to compare with AAV-packaged Cas13s in this manuscript. Please re-write the paragraph to provide reasoning for inducible systems, especially optical control. Deliverability was very well discussed in line 56. Maybe authors can move their discussion for compact Cas13 and AAV here.

We agree that a more refined explanation is warranted to delineate the connection between smaller-sized Cas13 systems and inducible platforms. As suggested, we rewrote the paragraph to clarify our meaning and address the reviewer’s concerns (**Page 3**).

3. Why was rapamycin-inducible dimerization explored? Was it chosen to inform optical control because it is a platform that allows for more high-throughput screening? Please provide more logic connection between chemogenetic control and optogenetic control.

We agree with the reviewer's point regarding the need for us to make a logical connection between chemogenetic and optogenetic control strategies. We selected rapamycin-based dimerization because of its well-established role in chemically controlled protein interactions. Notably, the chemical-based method offers more convenience in bulk-scale screening because there are few variables in the rapamycin treatment condition. For a light-inducible module, in contrast, the light stimulation must be optimized for the protein being conjugated to the optogenetic module. Based on our prior work, we optimized this optogenetic approach. As the reviewer suggested, we included this explanation in the revised manuscript (**Page 9**).

4. AF2 is not yet reliable in the prediction of RNP structures. RNA-guided RNA nucleases undergo major structural change upon acquisition of the crRNA (e.g., PMID 30917325, 35643083). The authors should acknowledge this in their explanation for split-design principles. Also, exactly how the 12 iterations of PspCas13 structure was done and what pLDDT was deemed satisfactory should be detailed in methods.

We acknowledge that AlphaFold2 is not yet fully reliable in predicting RNP structures, and RNA-guided RNA nucleases undergo significant structural changes upon acquiring the crRNA¹². However, we used the AlphaFold2 system for characterization, not to select split candidates. Our approach aimed to bridge the gap between experimental and computational screening. By combining AlphaFold2 analysis with experimental methods, we attempted to provide a better understanding of the auto-assembly behavior and functionality of our system. As the reviewer suggested, we included a statement that acknowledges this limitation and further clarifies our approach to ensure that readers of the revised manuscript fully understand our methodology (**Page 7 and Lines 121-125**).

Regarding the 12 iterations of the PspCas13b structure, the choice to increase the recycle number was influenced by the observation that the use of additional recycling (the default is 3) often enhances the accuracy of a structural model, as indicated in a paper that involved the use of ColabFold¹⁷. Given the lack of a well-defined three-dimensional structure for PspCas13b, we utilized 12 cycles to achieve a higher confidence in the predicted structure. We included these details in the revised manuscript (**Page 5**).

The pLDDT score is a per-residue confidence metric that is generated by AlphaFold2 and ranges from 0 to 100. This score estimates the confidence level for how well the predicted structure would align with an experimentally resolved structure. In our study, the average

pLDDT score for the final predicted structure of PspCas13b was 89.04, indicating a high degree of confidence in the predicted model. We included these details in the revised manuscript (**Methods, Pages 25-26**).

5. This reviewer believes that the targeting of 3'UTR was to accelerate luciferase mRNA degradation. However, the reason for this choice should be pointed out to the broad readership of Nature Communications.

We appreciate the reviewer's insightful comment about the choice to target the 3'UTR. We intended to obtain a clear representation of the knockdown effect while minimizing other variables that could complicate our interpretation. We chose to insert the $\lambda 2$ sequence in the 3'UTR because this sequence has a low Kd value¹⁸ and high efficiency in RNA knockdown and thus was expected to avoid confounding factors that could be introduced by targeting the 5'UTR or CDS, such as disruption of ribosomal function or translation. While our initial aim was to increase the clarity of interpretation and minimize variables, we agree with the reviewer's perspective. To address this, we revised to articulate a more straightforward reason for choosing the $\lambda 2$ sequence at the 3'UTR (**Page 6**).

6. Numbering of split sites follows two patterns (single number and N/C double numbers). Please specify their meanings as readers may not be well versed in split protein designs.

We greatly appreciate this valuable comment regarding the inconsistent numbering of split sites in the manuscript. We recognize the importance of clarity, especially for readers who may not be well-versed in split protein designs.

To address this, we standardized the numbering scheme to use the N/C double numbers across all figures and in the manuscript text. This format allows us to specify adjacent amino acids (e.g., N351/C352) and non-adjacent or overlapping ones (e.g., N351/C350). A clear explanation of this numbering scheme has been included in the Methods section (**Page 23**) to ensure that all readers can comprehend the text.

7. For spontaneous reconstitution of split Cas13 with split Fluc, was it guide-dependent? Please elaborate and provide information on what plasmids were transfected for each different experiment in experimental methods.

In response to this useful comment, we clarified the manuscript and figure legends to reflect that these experiments were conducted without crRNA to avoid any crRNA-dependent effects. The manuscript text detailing the experiments presented in new **Fig. 1f** has also been updated to include this information (**Method, Page 27, Lines 508-510**).

8. No negative controls for +488 nm light but –transfection of plasmids was included. This is a major issue with almost all the experiments in this manuscript. Short-wavelength light can induce the activation of multiple stress pathways as well as cross-talking cell growth pathways. A more solid negative control would be +488 nm light with the transfection of split Cas13 (without Magnet domains). Also, cell viability should be verified, too. The authors should seriously consider adding these controls before this manuscript can be published.

To address the reviewer’s suggestion that we use more robust negative controls, we included controls with “No transfection” and “Vehicle” to more specifically isolate the effects of light. We expanded our dataset to include evaluations of full-length and paCas13 (or padCas13 editor) under dark and light conditions (**Fig. R18a and b**, presented in new **Fig. 2g** and new **Supplementary Fig. 1**). To further ensure the robustness of our findings, we conducted cell viability tests for both active and inactive versions of Cas13 under various conditions, including “No transfection”, “Vehicle”, “Full-length”, and “Split” versions (**Fig. R18c and d**, presented in new **Fig. 3g** and new **Supplementary Fig. 1**).

The results from these additional experiments did not alter our interpretations, reinforcing the reliability of our original findings. These new data have been fully integrated into the revised manuscript and figures.

Again, we thank the reviewer for their invaluable feedback, which helped us significantly improve our manuscript.

Figure R18. Comprehensive controls for light-induced effects of paCas13 and padCas13 editor

a, Light-induced RNA knockdown of endogenous transcripts in HEK 293T cells transfected with full-length Cas13b or paCas13 candidates plus non-targeted (NT) or targeted (T) crRNAs. All data were normalized to the corresponding NT crRNA group within each dark and light condition. (n = 3 independent experiments)

b, Effect of light stimulation on RNA knockdown efficiency in HEK 293T cells transfected with full-length Cas13b or paCas13 candidates with targeted crRNAs. Data were normalized to corresponding groups in the dark condition. (n = 3 independent experiments)

c, Cell viability effects of transfected constructs related to active Cas13 under dark and light conditions. (n = 3 independent experiments)

d, Cell viability effects of transfected constructs related to inactive Cas13 under dark and light conditions. (n = 3 independent experiments)

9. Please provide potential explanations for nucleus localization of N-term fragment of Cas13. (Fig. 2d)

We thank the reviewer for their insightful comment regarding the nuclear localization of the N-terminal fragment of Cas13. We conducted additional experiments to provide a thorough explanation, such as by engaging in further cloning to enable immunofluorescence imaging and analyses. When we individually expressed each split candidate, we observed nuclear

localization patterns for the N286, N351, and N624 fragments (**Fig. R19**, presented in new **Supplementary Fig. 3**).

Firstly, we considered the intrinsic properties of Cas13. It has a high net charge of +14, which could potentially influence the localization of the split fragments, especially when certain domains are exposed by the splitting process. To clarify this, we added information about the electrostatic structure of PspCas13b to the revised manuscript (new **Supplementary Fig. 2d**).

Secondly, we focused on the inter-domain linker (IDL), since the PbuCas13b paper¹² describes it as having a high positive charge. The N286 fragment was the first in the sequence to display a nuclear localization pattern. The only difference between N286 and its preceding, non-nuclear-localized fragment, N272, was the inclusion of the IDL (**Fig. R19a and c**). This suggests that the IDL could be involved in directing the fragments to the nucleus.

Thirdly, we observed an intriguing pattern with fragment N624, which showed a strong nuclear localization (**Fig. R19c**) but, when co-transfected with its corresponding C-terminal fragment, was translocated entirely to the cytosol (**Fig. R19d**). The split-site N624/C625, which is rich in positively charged residues by crRNA pocket, also showed high efficiency in RNA knockdown due to auto-assembly (new **Fig. 2b**, new **Supplementary Fig. 2c and d**). Thus, it is possible that the electrostatic properties of the crRNA interaction region contribute to the behavior of N624.

In summary, we speculate that the nuclear localization of these fragments can be attributed to multiple factors, including **i**) the highly positive net charge of PspCas13; **ii**) the exposure of the IDL in N286 and N351; and **iii**) possibly the presence of a crRNA-interaction region in N624. These insights have been integrated into the revised manuscript to provide a more comprehensive understanding (**Fig. R19**, presented in new **Supplementary Fig. 3** and **Page 11**).

Figure R19. Subcellular localization of paCas13 fragments

a, Schematic of the construct designs for the paCas13 fragments.

b, Construction of HEPN domains with mCherry at the C-terminus. Representative 60X confocal microscopy images of HEK 293T cells are shown. Scale bar = 10 μ m.

c, Construction and expression of individual paCas13 fragments. Representative 60X confocal microscopy images of HEK 293T cells are shown. Scale bar = 10 μ m.

d, Co-expression of paCas13 fragments in the same well of a plate. Representative 60X confocal microscopy images of HEK 293T cells are shown. Scale bar = 10 μ m.

10. The paragraph between lines 190 and 206 should be rewritten to separate time-course study for activation and time-course study for reversibility. Right now Fig. 3c, e, f are for activation and 3d is for reversibility. For easier comprehension, please put luciferase activation and increase in corrected % by sequencing side-by-side. Then put luciferase decrease (current Fig 3d) and decrease in % editing by sequencing (authors should include this missing piece of data) side-by-side. Also, Fig. 3c says “normalized luciferase” for the y-axis. Please explain what value the timepoints were normalized to. This reviewer would assume timepoint 0 h, but it’s clearly not the case. The same problem is prevalent throughout the manuscript and should be amended, another example is Fig S5.

We agree with the reviewer that the arrangement of the figures could be improved to facilitate readers’ comprehension. To this end, we repositioned original **Fig. 3d** (new **Fig. 3h**) to focus on reversibility.

Regarding the “normalized luciferase” on the Y-axis, we understand the confusion this phrase might have caused. We revised the figures across the manuscript to specify that the values are normalized to Rluc per the dual-luciferase assay system. Additionally, we clarified the normalization standards for all relevant figures.

For original **Fig. 3c** (new **Fig. 3d**), we present the data as “Relative luminescence (fluc/Rluc)” without additional normalization to timepoint 0 h or dark conditions. This approach better shows the system’s inducibility and background activity and facilitates a direct comparison between padCas13-1 and padCas13-2.

In response to the reviewer’s request for additional data concerning reversibility, we included data obtained at additional time points. We conducted analyses every 3 hours within the specified interval (**Fig. R20a**, presented in new **Fig. 3h**). Our updated results revealed that padCas13-1 and padCas13-2 exhibited distinct kinetics, but shared a gradual and reversible transition to their original state.

We then conducted a direct analysis of RNA editing levels to obtain a more in-depth understanding of this reversibility. Unlike the luciferase signal, which peaked at the 12-hour mark before gradually decreasing, the RNA levels steadily declined to align with the dark state level by the 24-hour time point (**Fig. R20b**, presented in new **Fig. 3i**). Thus there were differences in the kinetics of RNA and protein synthesis and translation under the paCas13 condition.

The revised manuscript incorporates these additional data (**Fig. R20**, presented in new **Fig. 3h,i** and **Page 14**).

Figure R20 (Same as **Fig. R4**). Reversible optogenetic A-to-I RNA editing of the padCas13 editor

a, Reversibility of light-induced luciferase restoration by the padCas13 editor was assessed after blue-light illumination was removed. The data obtained at each time point were normalized to those obtained from the corresponding dark-exposed control group. (n = 3 independent experiments)

b, Direct quantification of RNA editing percentages was performed using RT-PCR-Sanger sequencing after blue-light illumination was removed. The data obtained at each time point were normalized to those obtained from the corresponding dark-exposed control group. (n = 3 independent experiments)

11. Please provide details on how mice bioluminescence data was quantified, i.e. selection of area of interest for quantification (superimposed onto all n = 5 mice images for each group).

To perform the quantification, we utilized Image Studio software (LI-COR) as follows.

We designated a region of interest (ROI) around the area exhibiting a bioluminescent signal for each analyzed luminescence signal. The ROI was chosen to cover the whole signal area; it was represented by a rectangle, the dimensions of which were kept consistent for all mice in each group to maintain uniformity. After setting the ROI, we analyzed the output from Image Studio, which provided us with the signal using the equation: $\text{Signal} = [\text{Total Intensity for ROI} - (\text{Background Mean Intensity} \times \text{Area (pixels) for ROI of interest})]$. These readings allowed us to compare the bioluminescent signals across mice of different groups. Finally, the gathered data were statistically analyzed using a t-test.

We ensured that the results were objective and comparable by consistently applying the same ROI dimensions and analytic method for all mice. As the reviewer suggested, we included details on our bioluminescence quantification in the revised manuscript (**Method, Page 31**).

12. Please provide images for all $n = 5$ mice for each group in Fig. S9a.

To address the reviewer's comment, we included images for all five mice of each group (new **Supplementary Fig. 9a**).

13. In the discussion section, the authors should provide more insights into the potential limitations of the paCas13 system and the challenges that need to be addressed in future studies. This will help readers better understand the scope of the study and its implications for future research.

In the revised manuscript, we expanded the Discussion section to address the key limitations and future challenges of the paCas13 system (**Pages 21-22**).

Other minor modifications include:

1. multiple typos and grammar errors throughout the manuscript, e.g., "CRIPSR" in line 25. These mistakes hinder the comprehension by readers and should be amended before publication.

We corrected this and invited native speakers to check the language throughout the revised manuscript.

References

1. Cox, D.B.T. et al. RNA editing with CRISPR-Cas13. *Science* **358**, 1019-1027 (2017).
2. Kawano, F., Okazaki, R., Yazawa, M. & Sato, M. A photoactivatable Cre-loxP recombination system for optogenetic genome engineering. *Nature Chemical Biology* **12**, 1059-1064 (2016).
3. Lee, J.-M. et al. Historical control data from 13-week repeated toxicity studies in Crj:CD (SD) rats. *lar* **28**, 115-121 (2012).
4. Meador, K. et al. Achieving tight control of a photoactivatable Cre recombinase gene switch: new design strategies and functional characterization in mammalian cells and rodent. *Nucleic Acids Research* **47**, e97-e97 (2019).
5. Morikawa, K. et al. Photoactivatable Cre recombinase 3.0 for in vivo mouse applications. *Nature Communications* **11**, 2141 (2020).
6. Yoshimi, K. et al. Photoactivatable Cre knock-in mice for spatiotemporal control of genetic engineering in vivo. *Laboratory Investigation* **101**, 125-135 (2021).
7. Li, H. et al. Stable Transgenic Mouse Strain with Enhanced Photoactivatable Cre Recombinase for Spatiotemporal Genome Manipulation. *Advanced Science* **9**, 2201352 (2022).
8. Dolberg, T.B. et al. Computation-guided optimization of split protein systems. *Nature Chemical Biology* **17**, 531-539 (2021).
9. Doman, J.L. et al. Phage-assisted evolution and protein engineering yield compact, efficient prime editors. *Cell* **186**, 3983-4002.e3926 (2023).
10. Abudayyeh, O.O. et al. A cytosine deaminase for programmable single-base RNA editing. *Science* **365**, 382-386 (2019).
11. Lee, S.-Y. et al. Engineered allostery in light-regulated LOV-Turbo enables precise spatiotemporal control of proximity labeling in living cells. *Nature Methods* **20**, 908-917 (2023).
12. Slaymaker, I.M. et al. High-Resolution Structure of Cas13b and Biochemical Characterization of RNA Targeting and Cleavage. *Cell Reports* **26**, 3741-3751.e3745 (2019).
13. Kannan, S. et al. Compact RNA editors with small Cas13 proteins. *Nature Biotechnology* (2021).
14. Xu, C. et al. Programmable RNA editing with compact CRISPR-Cas13 systems from uncultivated microbes. *Nature Methods* **18**, 499-506 (2021).
15. Charni-Natan, M. & Goldstein, I. Protocol for Primary Mouse Hepatocyte Isolation. *STAR Protocols* **1**, 100086 (2020).
16. Liu, F., Song, Y.K. & Liu, D. Hydrodynamics-based transfection in animals by systemic administration of plasmid DNA. *Gene Therapy* **6**, 1258-1266 (1999).
17. Mirdita, M. et al. ColabFold: making protein folding accessible to all. *Nature Methods* **19**, 679-682 (2022).
18. East-Seletsky, A. et al. Two distinct RNase activities of CRISPR-C2c2 enable guide-RNA processing and RNA detection. *Nature* **538**, 270-273 (2016).

REVIEWER COMMENTS

Reviewer #1 (Remarks to the Author):

This substantially revised manuscript shows notable improvement in addressing previously raised concerns. Specifically, the authors have presented compelling data demonstrating spatial and temporal control within living cells, expanded their inclusion of control experiments, and conducted more comprehensive comparisons involving various ADAR2 variants and Cas13-based editing systems. These efforts have significantly enhanced the overall quality of the manuscript.

However, a crucial aspect that requires further clarification and substantiation pertains to the method by which blue light can effectively reach engineered cells located deep within the liver. While it is unsurprising that blue light can modulate engineered devices positioned just beneath the skin (as the team has shown previously), given its maximal tissue penetration depth of about 1 mm, the challenge lies in explaining how this can be achieved within the intact liver in living animals, which typically resides at a considerably greater depth, ranging from 5 mm to 1 cm beneath the skin. This calculation considers the presence of subcutaneous fat tissue and other connective tissues beneath the dermis layer.

To address this significant concern and bolster the manuscript's scientific rigor, the authors are encouraged to provide a more detailed account of their light delivery methodology, and more solid dose-dependent data showing *in vivo* activation of the engineered devices. This should encompass essential parameters, such as the dose, frequency, power density, and wavelength of the blue light used in their experiments. Furthermore, a robust justification should be offered to elucidate how their chosen approach for light delivery can reliably reach and influence the engineered cells within the liver.

Reviewer #2 (Remarks to the Author):

The authors submitted a revised manuscript on their inducible Cas13 system. They have put in a substantial amount of work to address the reviewers' comments and I applaud them for their efforts. Their RNA engineering tool has two purported advantages over another recently published inducible Cas13b system (PMID: 36811465) that is based on similar ideas, namely (i) a better split site and (ii) the use of a safer inducer. Nevertheless, although their work is now much improved, the authors have run into a known challenge in the development of inducible CRISPR systems, which is a trade-off between unwanted background signal and desired switched-on activity. However, this is a difficult problem and there is no perfect solution to date. Overall, I think the authors have produced a reasonably good piece of work. Below are some comments that might improve the manuscript further.

Major comments:

- 1) Fig 2e: Please provide quantification of the percentages of cells localized in the nucleus vs cytoplasm when they are in the dark or exposed to light.
- 2) In Fig 3, please show the unscaled luminescence readings (firefly/renilla) for the original full-length enzymes, paCas13-1, and paCas13-2 at least for the 50nt spacer with 40nt mismatch distance.
- 3) Fig R2/R16 and Fig R3/R17 (in the rebuttal) should be presented in the main text, showing that N351 is a better split site than N761.
- 4) A key selling point of the current work is that blue light is a safer inducer than ABA-DMNB and UV light. However, Supplementary Fig 9 only contains data for paCas13 and does not show liver damage for the alternative system published in JACS (PMID: 36811465). The authors should perform a side-by-side comparison of their paCas13 system and the competing system under the same experimental conditions to demonstrate the safety advantage.

This is important because the original paper that reported the ABA chemically induced proximity (CIP) system (PMID: 21406691) claims that there is no known toxicity issues. Moreover, caged-ABA can be modified to respond to less hazardous light wavelengths. For example, ABA-DMNB (uncaged at 365nm) can be replaced with ABA-DEACM (uncaged at 405nm) (PMID: 25530501).

- 5) It would be good for the authors to demonstrate robustness of their paCas13 system. For example, besides testing multiple target loci, they can evaluate their technology in different cell lines. This can help convince a reader to try out their tool. Currently, as written, the reader might have some doubts on the efficacy of paCas13. For example, in Fig 2g, for paCas13, most do not show at least a 2-fold change in transcript levels. Additionally, in Fig 5, the paCas13 system does not compare well with the original full-length enzyme (paCas13 is not "within a range similar to that achieved with full-length Cas13", as claimed by the authors in the main text). More data are likely to be required to convince a reader to adopt paCas13.

Minor comments:

6) Introduction: The authors wrote "the combination of reduced size and light-inducible control enables our system to address the current limitations ..." However, they selected PspCas13b, which is one of the bigger Cas13 proteins.

7) Introduction: I suggest moving the sentence "Blue light is a less cytotoxic stimulus than UV light, making it safer and more broadly applicable for in vitro and in vivo applications" to the next paragraph, when the authors have mentioned their paCas13 system.

8) Fig 1f: It is unclear why only 3 split sites were tested here and how they were selected.

9) Fig 2f: Correlation between what? Please indicate in the figure legend.

10) Fig 3g: State duration of light treatment in the figure legend.

11) On page 14, the authors wrote "RNA levels aligning with the dark state ..." Should be "editing levels aligning with the dark state ..."

12) Fig 5d: Missing statistical tests.

Reviewer #3 (Remarks to the Author):

The authors of "Programmable RNA base editing with photoactivatable CRISPR-Cas13" have made substantial improvements to their manuscript in response to the feedback provided by peer reviewers. As such, I hereby extend my recommendation for the publication of this manuscript in Nature Communications.

Point-by-point responses to reviewers' comments

We are very pleased that Reviewer #3 is satisfied with our revisions. Thank you for the opportunity to respond to Reviewers #1 and 2. We are indebted to the reviewers for their careful reading of thoughtful comments on our work. Newly added or revised figures are summarized in the table below. The revised parts of the manuscript are indicated in red.

Figures newly added or revised.	
Main Fig. 2f	The nucleus-to-cytosol ratio for each fragment of the paCas13 system
Main Fig. 5d	Added statistical analysis by student t-test
Supplementary Fig. 3	Comparative analysis of RNA degradation under various induction conditions
Supplementary Fig. 7	Comparative comparison of A-to-I RNA base editing under various induction conditions
Supplementary Fig. 8	Optogenetic A-to-I RNA base editing of padCas13 editor in different cell lines
Supplementary Fig. 11	Blue-light penetration in a transverse mouse section
Supplementary Fig. 13	Light-dependent padCas13 editor activity in vivo and isolated mouse liver

Reviewer #1

We thank Reviewer #1 for recognizing the improvements in our revised manuscript and for providing additional constructive feedback.

Remarks to the Author:

This substantially revised manuscript shows notable improvement in addressing previously raised concerns. Specifically, the authors have presented compelling data demonstrating spatial and temporal control within living cells, expanded their inclusion of control experiments, and conducted more comprehensive comparisons involving various ADAR2 variants and Cas13-based editing systems. These efforts have significantly enhanced the overall quality of the manuscript.

However, a crucial aspect that requires further clarification and substantiation pertains to the method by which blue light can effectively reach engineered cells located deep within the liver. While it is unsurprising that blue light can modulate engineered devices positioned just beneath the skin (as the team has shown previously), given its maximal tissue penetration depth of about 1 mm, the challenge lies in explaining how this can be achieved within the intact liver in living animals, which typically resides at a considerably greater depth, ranging from 5 mm to 1 cm beneath the skin. This calculation considers the presence of subcutaneous fat tissue and other connective tissues beneath the dermis layer.

To address this significant concern and bolster the manuscript's scientific rigor, the authors are encouraged to provide a more detailed account of their light delivery methodology, and more solid dose-dependent data showing *in vivo* activation of the engineered devices. This should encompass essential parameters, such as the dose, frequency, power density, and wavelength of the blue light used in their experiments. Furthermore, a robust justification should be offered to elucidate how their chosen approach for light delivery can reliably reach and influence the engineered cells within the liver.

We acknowledge the Reviewer's concerns about the ability of blue light to effectively reach engineered cells located deep within the liver. However, the anatomy of a mouse differs significantly from that of a human. In mice, the skin is relatively thin and the liver is situated directly beneath the skin surface, especially in the abdominal region. Given this anatomical positioning, the mouse liver is more readily accessible to external stimuli, such as blue light. To clarify the anatomical distinction in mice, we now present a mouse tissue transverse section that clearly illustrates the proximity of the liver to the thin skin (**Fig. R1**, presented in new **Supplementary Fig. 11, Page 20, Lines 361-365**). This positioning in mice makes the liver readily accessible to blue light.

Figure R1. Blue light penetration in a transverse mouse section

a, Schematic illustration detailing the position from which we obtained the transverse section of the mouse (6-week-old Balb/c).

b, Anterior view of the dissected abdomen positioned under the fiber-type blue LED core ($\text{\O} 6$ mm, 470nm). Blue LED was delivered to the abdomen skin of the mouse.

c, Comparison of blue light penetration to the liver under varying light conditions.

Scale bar: 2 mm.

To further address the effectiveness of blue light in activating the engineered cells deep within the liver, we conducted additional experiments. We exposed mice to continuous blue light for 16 hours under varying intensities (Dark, 0.25, 0.5, 1, 5 mW/cm^2). This ensured that any observed luminescence signals would be attributable to the light intensity and not merely an accumulated effect over time. We observed a distinct increase in luminescence at 1 mW , indicating that blue light effectively activated our padCas13 system (**Fig. R2a and b**, presented in new **Supplementary Fig. 13**).

When we examined the livers of these mice, the luminescence was distinctly prominent along the edges of the liver, further confirming the influence of blue light on our system (**Fig. R2c and d**, presented in new **Supplementary Fig. 13**). This pattern contrasts with the general delivery of the reporter gene, which primarily showed signal along the blood vessels within the liver (**Fig. R3**, originally **Fig. R13 in the initial revision**). We have included these data and described the methodology in the revised manuscript (**Pages 20-21**).

Figure R2. Light-dependent padCas13 editor activity in vivo and isolated mouse livers

a, Intensity-dependent padCas13 editor activity measured from whole mice in vivo.

b, Luminescence images of the padCas13 editor system-carrying mice used to generate the data presented in panel a.

c, Intensity-dependent padCas13 editor activity of liver tissues isolated from the transfected mice shown in panel **b**.

d, Luminescence images of the isolated liver tissues used to generated the data presented in panel **c**.

Figure R3. Heterogeneous hepatocyte expression of the luciferase reporter

a, Luminescence images of mice bearing the luciferase reporter.

b, Luminescence images of whole livers collected from the mice shown in panel **a**.

c, Luminescence images of liver sections derived from the whole livers shown in panel **b**.

Reviewer #2

We thank Reviewer #2 for recognizing our efforts in revising the manuscript and for providing further constructive feedback.

Remarks to the Author:

The authors submitted a revised manuscript on their inducible Cas13 system. They have put in a substantial amount of work to address the reviewers' comments and I applaud them for their efforts. Their RNA engineering tool has two purported advantages over another recently published inducible Cas13b system (PMID: 36811465) that is based on similar ideas, namely (i) a better split site and (ii) the use of a safer inducer. Nevertheless, although their work is now much improved, the authors have run into a known challenge in the development of inducible CRISPR systems, which is a trade-off between unwanted background signal and desired switched-on activity. However, this is a difficult problem and there is no perfect solution to date. Overall, I think the authors have produced a reasonably good piece of work. Below are some comments that might improve the manuscript further.

Major comments:

1) Fig 2e: Please provide quantification of the percentages of cells localized in the nucleus vs cytoplasm when they are in the dark or exposed to light.

Due to the heterogenous localization of the N-terminal fragment, it was not feasible to subjectively determine the percentages of cells with signal in the nucleus vs cytosol. Therefore, we adopted a quantitative approach by measuring the nucleus-to-cytosol ratio normalized against BFP expression. Building on the colocalization data presented in our initial revision using Pearson's correlation coefficient (**Fig. 2f**, presented in new **Fig. 2g**), we updated Figure 2e in the revised manuscript to include these quantified data (new **Fig. 2f-g**).

2) In Fig 3, please show the unscaled luminescence readings (firefly/renilla) for the original full-length enzymes, paCas13-1, and paCas13-2 at least for the 50nt spacer with 40nt mismatch distance.

We appreciate the Reviewer's comment on the data presented in Figure 3c. We would like to clarify that the luminescence readings shown are already the unscaled, raw Firefly luminescence readings normalized to those of Renilla (internal control). We used Renilla

normalization to account for well-to-well variations and ensure a more accurate data representation.

3) Fig R2/R16 and Fig R3/R17 (in the rebuttal) should be presented in the main text, showing that N351 is a better split site than N761.

We understand the importance of emphasizing the efficacy of the 351 split site. In response to the Reviewer's comment, we now include a comparison between the 351 and 761 split sites in the revised manuscript (new **Supplementary Fig. 3 and 7** and **Pages 11-12 and 17**).

4) A key selling point of the current work is that blue light is a safer inducer than ABA-DMNB and UV light. However, Supplementary Fig 9 only contains data for paCas13 and does not show liver damage for the alternative system published in JACS (PMID: 36811465). The authors should perform a side-by-side comparison of their paCas13 system and the competing system under the same experimental conditions to demonstrate the safety advantage.

This is important because the original paper that reported the ABA chemically induced proximity (CIP) system (PMID: 21406691) claims that there is no known toxicity issues. Moreover, caged-ABA can be modified to respond to less hazardous light wavelengths. For example, ABA-DMNB (uncaged at 365nm) can be replaced with ABA-DEACM (uncaged at 405nm) (PMID: 25530501).

We appreciate the Reviewer's valuable feedback regarding the safety comparison of our blue light-induced system. Upon careful consideration and after discussion with the editor, we believe that a direct experimental comparison might not be necessary for the following reasons:

1. The published JACS paper does not provide in vivo data generated using UV light; therefore, a direct comparison with our system is not feasible.
2. Our current lab setup has limitations in terms of conducting UV light experiments, especially in vivo.
3. Direct comparison is further complicated by the absence of in vivo data for the photocaged-ABA system.
4. We previously conducted a thorough comparison of inducible systems in our initial revision response.
5. The primary innovation of our work is the application of blue light and the split site for spatiotemporal regulation.

Given these considerations and the editor's guidance that our existing data sufficiently demonstrates our system's safety, we instead added a more in-depth textual comparison in our revised manuscript (Pages 11-12 and 17).

5) It would be good for the authors to demonstrate robustness of their paCas13 system. For example, besides testing multiple target loci, they can evaluate their technology in different cell lines. This can help convince a reader to try out their tool. Currently, as written, the reader might have some doubts on the efficacy of paCas13. For example, in Fig 2g, for paCas13, most do not show at least a 2-fold change in transcript levels. Additionally, in Fig 5, the paCas13 system does not compare well with the original full-length enzyme (paCas13 is not "within a range similar to that achieved with full-length Cas13", as claimed by the authors in the main text). More data are likely to be required to convince a reader to adopt paCas13.

We appreciate the Reviewer's suggestion that we demonstrate the robustness of our system. To address this point, we expanded our work to include additional cell lines representing different tissue types and species, namely the HeLa (human cervical carcinoma), MCF7 (human breast cancer), HT1080 (human fibrosarcoma), and Neuro-2a (mouse neuroblasts) cell lines. Our findings, which we have included in the revised manuscript (Fig. R4, presented in new Supplementary Fig. 8 and Page 17), show that the padCas13 editor yields effective light-inducible base editing in these diverse cell lines, thereby reinforcing the potential utility of our system for various research applications.

Figure R4. Optogenetic A-to-I RNA base editing by the padCas13 editor in various cell lines

HeLa, MCF7, HT1080, and Neuro-2a cells were transfected with a full-length Cas13b or the padCas13-2 editor system with non-targeted (NT) or targeted crRNAs. All values were normalized to the corresponding NT crRNA group within each dark or light condition. (n = 3 independent experiments)

Minor comments:

6) Introduction: The authors wrote "the combination of reduced size and light-inducible control enables our system to address the current limitations ..." However, they selected PspCas13b, which is one of the bigger Cas13 proteins.

In our introduction, "reduced size" pertains to the strategic splitting of the Cas13 protein, which allows the use of a smaller delivery vector than that required by the full-length system. We have clarified this point in the revised manuscript to reflect better the advantages of our system (**Page 4, Lines 51-52**).

7) Introduction: I suggest moving the sentence "Blue light is a less cytotoxic stimulus than UV light, making it safer and more broadly applicable for in vitro and in vivo applications" to the next paragraph, when the authors have mentioned their paCas13 system.

We appreciate the Reviewer's suggestion to refine the introduction. We intended to establish the context of current limitations before introducing our paCas13 system. However, we understand the Reviewer's point and have adjusted the manuscript to immediately emphasize the relevance of blue light when introducing our paCas13 system (**Page 4, Lines 66-67**).

8) Fig 1f: It is unclear why only 3 split sites were tested here and how they were selected.

During our rapamycin screening process, we chose three distinct sites for testing based on their varied RNA degradation patterns: one that exhibited no inducibility, another that displayed inducibility, and a third that demonstrated the highest level of auto-assembly. To clarify this, we have included in the revised manuscript a description of our screening process and rationale for selecting these specific sites (**Page 8**).

9) Fig 2f: Correlation between what? Please indicate in the figure legend.

In Figure 2f, we are referring to the Pearson's correlation coefficient presented in Figure 2e, which quantifies the colocalization of the N- and C-terminal fragments of the paCas13 system. We have revised the figure legend to explicitly state this, thereby ensuring that the correlation is clearly defined for the reader (original **Fig. 2f**, presented in new **Fig. 2g, Pages 32-33 in Methods**).

10) Fig 3g: State duration of light treatment in the figure legend.

While all blue light-exposure conditions were consistent and detailed in the Methods section, we inadvertently omitted the specific duration of light treatment for the cell viability assays. To ensure clear and unambiguous data representation, we added this information to the Figure legends and Methods section in the revised manuscript (**Page 30 in Methods**).

11) On page 14, the authors wrote "RNA levels aligning with the dark state ..." Should be "editing levels aligning with the dark state ..."

We corrected the wording to “editing levels” on **Page 16**.

12) Fig 5d: Missing statistical tests.

To address the Reviewer’s comment, we included statistical information in **Figure 5d**.

Reviewer #3

Remarks to the Author:

The authors of "Programmable RNA base editing with photoactivatable CRISPR-Cas13" have made substantial improvements to their manuscript in response to the feedback provided by peer reviewers. As such, I hereby extend my recommendation for the publication of this manuscript in Nature Communications.

We are immensely grateful that Reviewer #3 is satisfied with our revisions. The Reviewer's constructive comments have been invaluable in this process. Thank you once again for your support and recommendation.

REVIEWERS' COMMENTS

Reviewer #1 (Remarks to the Author):

The authors have addressed my previous comments by performing a dose-dependent assay and providing additional data to demonstrate the efficacy of light delivery to the subcutaneous tissue. I commend the authors for their thorough work. I have no remaining concerns and I recommend this manuscript for publication.

Reviewer #2 (Remarks to the Author):

The authors have made reasonable efforts to address the remaining reviewers' concerns. Hence, I recommend publication in Nature Communications.